# ON THE POSSIBILITIES OF AI-GENERATED TEXT DETECTION: A SAMPLE COMPLEXITY ANALYSIS

## ABSTRACT

Our work addresses the critical issue of distinguishing text generated by Large Language Models (LLMs) from human-produced text, a task essential for numerous applications. Despite ongoing debate about the feasibility of such differentiation, we present evidence supporting its consistent achievability, except when human and machine text distributions are indistinguishable across their entire support. Drawing from information theory, we argue that as machine-generated text approximates human-like quality, the sample size needed for detection increases. We establish precise sample complexity bounds for detecting AI-generated text, laying groundwork for future research aimed at developing advanced, multi-sample detectors. Our empirical evaluations across multiple datasets (Xsum, Squad, IMDb, and Kaggle FakeNews) confirm the viability of enhanced detection methods. We test various state-of-the-art text generators, including GPT-2, GPT-3.5-Turbo, Llama, Llama-2-13B-Chat-HF, and Llama-2-70B-Chat-HF, against detectors, including oBERTa-Large/Base-Detector, GPTZero. Our findings align with OpenAI's empirical data related to sequence length, marking the first theoretical substantiation for these observations.

## 1 INTRODUCTION

Large Language Models (LLMs) like GPT-3 mark a significant milestone in the field of Natural Language Processing (NLP). Pre-trained on vast text corpora, these models excel in generating contextually relevant and fluent text, advancing a variety of NLP tasks including language translation, question-answering, and text classification. Notably, their capacity for zero-shot generalization obviates the need for extensive task-specific training. Recent research by (Shin et al., 2021) further highlights the LLMs' versatility in generating diverse writing styles, ranging from academic to creative, without the need for domain-specific training. This adaptability extends their applicability to various use-cases, including chatbots, virtual assistants, and automated content generation.

However, the advanced capabilities of LLMs come with ethical challenges (Bommasani et al., 2022). Their aptitude for generating coherent, contextually relevant text opens the door for misuse, such as the dissemination of fake news and misinformation. These risks erode public trust and distort societal perceptions. Additional concerns include plagiarism, intellectual property theft, and the generation of deceptive product reviews, which negatively impact both consumers and businesses. LLMs also have the potential to manipulate web content maliciously, influencing public opinion and political discourse.

Given these ethical concerns, there is an imperative for the responsible development and deployment of LLMs. The ethical landscape associated with these models is complex and multifaceted. Addressing these challenges is vital for harnessing the societal benefits that responsibly deployed LLMs can offer. To this end, recent research has pivoted towards creating detectors capable of distinguishing text generated by machines from that authored by humans. These detectors serve as a safeguard against the potential misuse of LLMs. One central question underpinning this area of research is:

*"Is it possible to detect the AI-generated text in practice?"*

Our work provides an affirmative answer to this question. Specifically, we demonstrate that detecting AI-generated text is nearly always feasible, provided multiple samples are collected, as illustrated in Figure 1. The necessity for collecting multiple samples is consistent with real-world settings where

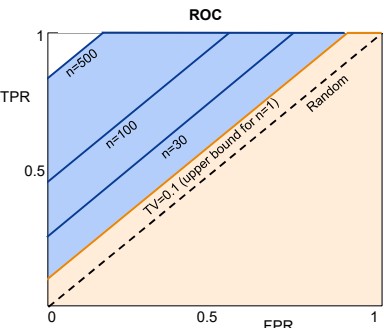 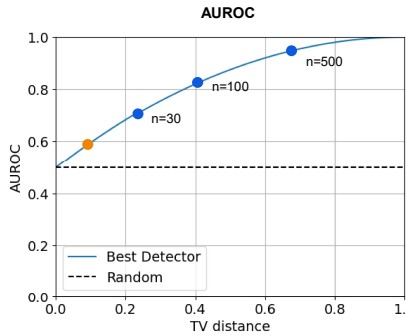

Figure 1: In light of the sample complexity bound presented in Theorem 1, we show here pictorially how increasing the number of samples $n$ used for detection would affect the ROC of the best possible detector, which is achieved by the likelihood-ratio-based classifier. We note that in the ROC curve on the left for $\text{TV}(m, h) = 0.1$, the AUROC of the best possible detector will be $0.6$ as derived in (Sadasivan et al., 2023) (shown by an orange dot in right figure). The AUROC of $0.6$ would lead to the conclusion that detection is hard. In contrast, we note that by increasing the number of samples $n$, the ROC upper bound starts increasing towards $1$ exponentially fast (shown by the shaded blue region in the left figure for different values of $n$), and hence the AUROC of the best possible detector also starts increasing as shown by corresponding blue dots in the right figure. This ensures that the detection should be possible even in hard scenarios when $\text{TV}(m, h)$ norm is small.

data is abundant. For example, in the context of social media bot identification, one can readily collect multiple posts to determine their origin, whether machine-generated or human-authored. This real-world applicability emphasizes the importance and urgency of developing sophisticated detection mechanisms for the ethical use of LLMs. We summarize our main contributions as follows.

(1) **Possibility of AI-generated text detection**: We utilize a mathematically rigorous approach to answer the question of the possibility of AI-generated text detection. We conclude that there is a *hidden possibility* of detecting the AI text, which improves with the text sequence (token) length.

(2) **Sample complexity of AI-generated text detection**: We derive the sample complexity bounds, a first-of-its-kind tailored for detecting AI-generated text for both IID and non-IID settings.

(3) **Comprehensive Empirical Evaluations:** We have conducted extensive empirical evaluations for real datasets Xsum, Squad, IMDb, and Fake News dataset with state-of-the-art generators (GPT-2, GPT3.5 Turbo, Llama, Llama-2 (13B), Llama-2 (70B)) and detectors (OpenAI's Roberta (large), OpenAI's Roberta (base), and ZeroGPT (SOTA Detector)).

## 2 BACKGROUND ON AI-GENERATED TEXT DETECTORS AND RELATED WORKS

Recent research has shown promising results in developing detection methods. Some of these methods use statistical approaches to identify differences in the linguistic patterns of human and machine-generated text. We survey the existing approaches here.

**Traditional approaches.** They involve statistical outlier detection methods, which employ statistical metrics such as entropy, perplexity, and $n$-gram frequency to differentiate between human and machine-generated texts (Lavergne et al., 2008; Gehrmann et al., 2019). However, with the advent of ChatGPT (OpenAI) (OpenAI, 2023), a new innovative statistical detection methodology, DetectGPT (Mitchell et al., 2023), has been developed. It operates on the principle that text generated by the model tends to lie in the negative curvature areas of the model's log probability. DetectGPT (Mitchell et al., 2023) generates and compares multiple perturbations of model-generated text to determine whether the text is machine-generated or not based on the log probability of the original text and the perturbed versions. DetectGPT significantly outperforms the majority of the existing zero-shot methods for model sample detection with very high AUC scores (note that we use the terms AUROC and AUC interchangeably for presentation convenience).

**Classifier-based detectors.** In contrast to statistical methods, classifier-based detectors are common in natural language detection paradigms, particularly in fake news and misinformation detection

(Schildhauer, 2022; Zou & Ling, 2021). OpenAI has recently fine-tuned a GPT model (OpenAI, 2023) using data from Wikipedia, WebText, and internal human demonstration data to create a web interface for a discrimination task using text generated by $34$ language models. This approach combines a classifier-based approach with a human evaluation component to determine whether a given text was machine-generated or not. These recent advancements in the field of detecting AI-generated text have significant implications for detecting and preventing the spread of misinformation and fake news, thereby contributing to the betterment of society (Schildhauer, 2022; Zou & Ling, 2021; Kshetri & Voas, 2022).

**Watermark-based identification.** An alternative detection paradigm that has garnered significant interest in this field is the evolution of watermark-based identification (Verma et al., 2009; Wadhera et al., 2022). One of the most exciting works in recent times around this research revolves around watermarking and developing efficient watermarks for machine-generated text detection. Historically, watermarks have been employed in the realm of image processing and computer vision to safeguard copyrighted content and prevent intellectual property theft (Langelaar et al., 2000). They can also be used for data hiding, where information is hidden within the watermark itself, allowing for secure and discreet transmission of information. Early research by (Atallah et al., 2001; Meral et al., 2009) was among the first to demonstrate the potential of watermarks in language through syntax tree manipulations. More recently with the advent of ChatGPT, innovative work by (Kirchenbauer et al., 2023) has shown how to incorporate watermarks by using only the LLM's logits at each step. The watermarking technique proposed by (Kirchenbauer et al., 2023) allows for the verification of a watermark's authenticity by employing a specific hash function. More specifically, the soft watermarking approach by (Kirchenbauer et al., 2023) involves categorizing tokens into "green" and "red" lists for generating distinct patterns. Watermarked language models are more likely to select tokens from the green list, based on prior tokens, resulting in watermarks that are typically unnoticeable to humans. These advancements in watermarking technology not only strengthen copyright protection and content authentication but also open up new avenues for research in areas such as privacy in language, secure communication, and digital rights management.

**Impossibility result**. The interesting recent literature by (Sadasivan et al., 2023; Krishna et al., 2023) showed the vulnerabilities of watermark-based detection methodologies using vanilla paraphrasing attacks. (Sadasivan et al., 2023) developed a lightweight neural network-based paraphraser and applied it to the output text of the AI-generative model to evade a whole range of detectors, including watermarking schemes, neural network-based detectors, and zero-shot classifiers. (Sadasivan et al., 2023) also introduced a notion of spoofing attacks where they exposed the vulnerability of LLMs protected by watermarking under such attacks. (Krishna et al., 2023) on the other hand, trained a paraphrase generation model capable of paraphrasing paragraphs and showed that paraphrased texts with DIPPER (Krishna et al., 2023) evade several detectors, including watermarking, GPTZero, DetectGPT, and OpenAI's text classifier with a significant drop in accuracy. Additionally, (Sadasivan et al., 2023) highlighted the impossibility of machine-generated text detection when the total variation (TV) norm between human and machine-generated text distributions is small.

In this work, we show that there is a ***hidden possibility*** of detecting the AI-generated text even if the TV norm between human and machine-generated text distributions is small. This result is in support of the recent detection possibility claims by Krishna et al. (2023).

## 3    PROPOSED APPROACH: METHODOLOGY AND ANALYSIS

### 3.1    NOTATIONS AND DEFINITIONS

Before discussing the main results, let us define the notations used in this paper. We define the set of all possible texts (textual representations) as $\mathcal{S}$, a human-generated text distribution as $h(s)$ over $s \in \mathcal{S}$, and machine-generated text distribution as $m(s)$. Here $m(s)$ and $h(s)$ are valid probability density functions. We can also modify the same notation given a specific prompt (noted by $p$) or context (denoted by $c$) or question (denoted by $q$) accordingly, such as $\mathcal{S}_c$, $h(s \mid p, c, q)$, and $m(s \mid p, c, q)$ respectively. However, for the sake of clarity and ease of discussion in this work, we will omit the use of complex notation.

In the literature, the problem of detecting AI-generated text is considered as a binary classification problem. The (potentially nonlinear and complex) detector $D(s)$ maps the sample $s \in \mathcal{S}$ to $\mathbb{R}$

for possible binary classification, and then compares it against a threshold $\gamma$ to perform detection. $D(s) \geq \gamma$ is classified as AI-generated while $D(s) < \gamma$ is categorized as human-generated. For the detector $D(s)$ to detect whether the text samples $s$ is generated from the machine or not, we need to study the receiver operating characteristic curve (ROC curve) (Fawcett, 2006), which involves two terms, namely True Positive Rate (TPR) and False Positive Rate (FPR). Once we obtain ROC, we can study the area under the ROC curve AUROC, which characterizes the detection performance of detector $D$. The upper bound on AUROC describes the performance of the best possible detector.

Under a detection threshold $\gamma$, TPR and FPR are denoted as $\text{TPR}_\gamma$ and $\text{FPR}_\gamma$ respectively:

$$\text{TPR}_\gamma : \text{Probability of detecting AI-generated text as AI-generated under threshold } \gamma, \tag{1}$$

$$\text{FPR}_\gamma : \text{Probability of detecting human-generated text as AI-generated under threshold } \gamma. \tag{2}$$

The rigorous definitions of $\text{TPR}_\gamma$ and $\text{FPR}_\gamma$ are as follows.

$$\text{TPR}_\gamma = \mathbb{P}_{s \sim m(\cdot)}[D(s) \geq \gamma] = \int \mathbb{I}_{\{D(s) \geq \gamma\}} \cdot m(s) \cdot ds, \tag{3}$$

$$\text{FPR}_\gamma = \mathbb{P}_{s \sim h(\cdot)}[D(s) \geq \gamma] = \int \mathbb{I}_{\{D(s) \geq \gamma\}} \cdot h(s) \cdot ds, \tag{4}$$

where $\mathbb{I}_{\{\text{condition}\}}$ is the indicator function which takes value 1 if the condition is true, and 0 otherwise. Note that without loss of generality, we have chosen to consider $m(s)$ and $h(s)$ as the probability density function of machine and human on a sample $s$ by considering continuous $s$ (as also considered in (Sadasivan et al., 2023)), but similar results hold for discrete $s$ by replacing the integral with a summation and by considering $m(s)$ and $h(s)$ as the probability mass function of machine and human on a sample $s$.

Both $\text{TPR}_\gamma$ and $\text{FPR}_\gamma$ are within the closed interval [0, 1] for any threshold $\gamma$. For a good detector, $\text{TPR}_\gamma$ should be as high as possible, and $\text{FPR}_\gamma$ should be as low as possible. As a result, a high *area under the ROC curve* (AUROC) is desirable for detection. AUROC is between 1/2 and 1, i.e., $\text{AUROC} \in [1/2, 1]$. An AUROC value of $1/2$ means a random detection and a value of 1 indicates a perfect detection. For efficient detection, the goal is to design a detector $D$ such that AUROC is as high as possible.

### 3.2 Hidden Possibilities of AI-Generated Text Detection

To study the AUROC for any detector $D$, we start by invoking LeCam's lemma (Le Cam, 2012; Wasserman, 2013) which states that for any distributions $m$ and $h$, given an observation $s$, the minimum sum of Type-I and Type-II error probabilities in testing whether $s \sim m$ versus $s \sim h$ is equal to $1 - \text{TV}(m, h)$. Hence, mathematically, we can write

$$\underbrace{\mathbb{P}_{s \sim h(\cdot)}[D(s) \geq \gamma]}_{\text{Type-I error (false positive)}} + \underbrace{\mathbb{P}_{s \sim m(\cdot)}[D(s) < \gamma]}_{\text{Type-II error (false negative)}} \geq 1 - \text{TV}(m, h), \tag{5}$$

for any detector $D$ and any threshold $\gamma$. We note that the above bound is tight and can always be achieved with equality by likelihood-ratio-based detectors for *any* distribution $m$ and $h$, by the Neyman-Pearson Lemma (Cover, 1999, Chapter 11). We restate the lemma for completeness and discuss its tightness in Appendix B.1. From the definitions of TPR and FPR in (3)-(4), it holds that

$$\text{FPR}_\gamma + 1 - \text{TPR}_\gamma \geq 1 - \text{TV}(m, h), \tag{6}$$

which implies that

$$\text{TPR}_\gamma \leq \min\{\text{FPR}_\gamma + \text{TV}(m, h), 1\}, \tag{7}$$

where min is used because $\text{TPR}_\gamma \in [0, 1]$. The upper bound in (7) is called the ROC upper bound and is the bound leveraged in one of the recent works (Sadasivan et al., 2023) to derive AUROC upper bound $\text{AUC} \leq \frac{1}{2} + \text{TV}(m, h) - \frac{\text{TV}(m,h)^2}{2}$ which holds for any $D$. This upper bound led to the claim of the impossibility of detecting the AI-generated text whenever $\text{TV}(m, h)$ is small.

**Hidden Possibility.** However, we note that the claim of impossibility from the AUROC upper bound could be too conservative for detection in practical scenarios. For instance, we provide a ***motivating example*** of detecting whether an account on Twitter is an AI-bot or human. It is natural that we will

have a collection of text samples from the account, denoted by $\{s_i\}_{i=1}^n$, and it is realistic to assume that $n$ is very high. Therefore, the natural practical question is whether we can detect if the provided text set $\{s_i\}_{i=1}^n$ is machine-generated or human-generated. With this motivation, we next explain that detection is always possible.

We formalize the problem setting and prove our claim by utilizing the existing results in the information theory literature. Let us consider the same setup as detailed before, while we are given a set of samples $S := \{s_i\}_{i=1}^n$. For simplicity, we assume that the samples are i.i.d. drawn from either the human $h$ or machine $m$. Interestingly, now the hypothesis test can be re-written as

$$H_0 : S \sim m^{\otimes n} \quad \text{v.s.} \quad H_1 : S \sim h^{\otimes n}, \tag{8}$$

where $m^{\otimes n} := m \otimes m \otimes \cdots \otimes m$ ($n$ times) denotes the product distribution, as does $h^{\otimes n}$. This is one of the key observations that focus on the correct hypothesis-testing framework with multiple samples. Similar to before (cf. 7), based on Le Cam's lemma, it holds that now $1 - \mathsf{TV}(m^{\otimes n}, h^{\otimes n})$ gives the minimum Type-I and Type-II error rate, which implies

$$\text{TPR}_\gamma^n \leq \min\{\text{FPR}_\gamma^n + \mathsf{TV}(m^{\otimes n}, h^{\otimes n}), 1\}, \tag{9}$$

where

$$\text{TPR}_\gamma^n = \mathbb{P}_{S \sim m^{\otimes n}}[D(S) \geq \gamma] = \int \mathbb{I}_{\{D(S) \geq \gamma\}} \cdot m^{\otimes n}(S) \cdot dS, \tag{10}$$

$$\text{FPR}_\gamma^n = \mathbb{P}_{S \sim h^{\otimes n}}[D(S) \geq \gamma] = \int \mathbb{I}_{\{D(S) \geq \gamma\}} \cdot h^{\otimes n}(S) \cdot dS. \tag{11}$$

We emphasize that the term $\mathsf{TV}(m^{\otimes n}, h^{\otimes n})$ is an increasing sequence in $n$ and eventually converges to 1 as $n \to \infty$. Due to the data processing inequality, it holds that $\mathsf{TV}(m^{\otimes k}, h^{\otimes k}) \leq \mathsf{TV}(m^{\otimes n}, h^{\otimes n})$ when $k \leq n$ and naturally leads to $\mathsf{TV}(m, h) \leq \mathsf{TV}(m^{\otimes n}, h^{\otimes n})$. This is a crucial observation, showing that even if the machine and human distributions were close in the sentence space, by collecting more sentences, it is possible to inflate the total variation norm to make the detection possible.

Now, from the large deviation theory, we can show that the rate at which total variation distance approaches 1 is exponential with the number of samples (Polyanskiy & Wu, 2022, Chapter 7),

$$\mathsf{TV}(m^{\otimes n}, h^{\otimes n}) = 1 - \exp\left(-nI_c(m, h) + o(n)\right), \tag{12}$$

where, $I_c(m, h)$ is known as the *Chernoff information* and is given by $I_c(m, h) = -\log \inf_{0 \leq \alpha \leq 1} \int m^\alpha(s) h^{1-\alpha}(s) ds$. The above expressions lead to Proposition 1 next.

**Proposition 1** (**Area Under ROC Curve**). *For any detector $D$, with a given collection of i.i.d. samples $S := \{s_i\}_{i=1}^n$ either from human $h(s)$ or machine $m(s)$, it holds that*

$$\textsf{AUROC} \leq \frac{1}{2} + \mathsf{TV}(m^{\otimes n}, h^{\otimes n}) - \frac{\mathsf{TV}(m^{\otimes n}, h^{\otimes n})^2}{2}, \tag{13}$$

*where $\mathsf{TV}(m^{\otimes n}, h^{\otimes n}) := 1 - \exp\left(-nI_c(m, h) + o(n)\right)$ and $I_c(m, h)$ is the Chernoff information. Therefore, the upper bound of $\textsf{AUROC}$ increases exponentially with respect to the number of samples $n$.*

The proof of the above proposition follows by integrating the $\text{TPR}_\gamma^n$ upper bound in (9) over $\text{FPR}_\gamma^n$. We note that the expression in (13) and the equality of $\mathsf{TV}$ distance in terms of Chernoff information presents an interesting connection between the number of samples and $\textsf{AUROC}$ of the best possible detector (which archives the bound in (9) with equality). It is evident that if we increase the number of samples, $n \to \infty$, the total variation distance $\mathsf{TV}(m^{\otimes n}, h^{\otimes n})$ approaches 1 and that too exponentially fast, and hence increasing the $\textsf{AUROC}$. This indicates that as long as the two distributions are not exactly the same, which is rarely the same, the detection will always be possible by collecting more samples as established next.

### 3.3 ATTAINABILITY OF THE AUROC UPPER-BOUND VIA LIKELIHOOD-RATIO-BASED DETECTORS

**Likelihood-ratio-based Detector.** Here, we discuss the attainability of bounds in Proposition 1 to establish that the bound is indeed tight. We note that it is a well-established fact in the literature that

a likelihood-ratio-based detector would attain the bound for any distributions $h$ and $m$ and hence is the best possible detector (detailed proof provided in Appendix B.1). We discuss the likelihood-ratio-based detector here for completeness in the context of LLMs as follows. Specifically, the likelihood ratio-based detector is given by

$$D^*(S) := \begin{cases} \text{Text from machine} & \text{if } m^{\otimes n}(S) \geq h^{\otimes n}(S), \\ \text{Text from human} & \text{if } m^{\otimes n}(S) < h^{\otimes n}(S). \end{cases} \tag{14}$$

We proved in Appendix B.1 that the detector in (14) attains the bound and is the best possible detector.

**Sample Complexity of Best Possible Detector.** To further emphasize the dependence on the number of samples $n$, we derive the sample complexity bound of AI-generated text detection in Theorem 1 as follows.

**Theorem 1 (Sample Complexity of AI-generated Text Detection (Possibility Result)).** *If human and machine distributions are close* $\mathsf{TV}(m, h) = \delta > 0$*, then to achieve an* AUROC *of* $\epsilon$*, we require*

$$n = \Omega \left( \frac{1}{\delta^2} \log \left( \frac{1}{1 - \epsilon} \right) \right) \tag{15}$$

*number of samples for the best possible detector which is likelihood-ratio-based as mentioned in (14), for any* $\epsilon \in [0.5, 1)$*. Therefore, AI-generated text detection is possible for any* $\delta > 0$*.*

The proof of Theorem 1 is provided in Appendix B.2. From the statement of Theorem 1, it is clear that, as long as $\delta > 0$ (which means no matter how close human $h(s)$ and $m(s)$ distributions are) and $\epsilon < 1$, there exists $n$ such that we can achieve high AUROC and perform the detection. Here, $n$ corresponds to the number of sentences generated by either humans or machines which we need to detect. We provide additional detailed remarks and insights in Appendix A.

## 3.4 EXTENSION TO NON-IID CASE

We extend the sample complexity results of Theorem 1 to the non-iid setting in this subsection. In order to accomplish that, we make certain assumptions about the structures present in the input, which is a well-founded assumption that proves to be practical and applicable in the context of various natural language tasks (for ex: present of topics in documents (Jelodar et al., 2018; Loureiro et al., 2023)). Let us denote the strength of the association with $\rho$ to characterize the dependence between the sequences $s_i$ and the dependence is given as

$$\mathbb{E}[S_i | S_{i-1} = s_{i-1}, \cdots, S_1 = s_1] = \rho \frac{\sum_{k=1}^{i-1} s_k}{i - 1} + (1 - \rho)\mathbb{E}[S_i], \tag{16}$$

which boils down to the iid case for $\rho = 0$. An increasing $\rho$ indicates increasing dependence on the previous sequence with $\rho = 1$ means that the conditional expectation can be completely expressed in terms of the previous samples in the sequence. The dependence assumption of (16) embodies a natural intuition for the domain of natural language and serves as a foundation for extending our results to non-iid scenarios. Eq. (16) provides a way to measure the dependence between random variables, which is later used to extend Chernoff bound to non-iid cases. . In the context of LLMs, one can think of the sum $\sum s_k$ as the "average meaning" of these text samples, such as "woman" + "royalty" may have a similar meaning as "queen".

Before introducing the final result, let us assume the number of sequences or samples is denoted by $n$, there are $L$ independent subsets, and the corresponding subset is represented by $\tau_j$ where $j \in (1, 2 \cdots, L)$, where $\tau_j$ consists of $c_j$ samples (dependent). This is a natural assumption in NLP where a large paragraph often consists of multiple topics, and sentences for each topic are dependent. With the above definitions, we state the main result in Theorem 2 for the non-iid setting.

**Theorem 2 (Sample Complexity of AI-generated Text Detection (non-iid)).** *If human and machine distributions are close* $\mathsf{TV}(m, h) = \delta > 0$*, then to achieve an* AUROC *of* $\epsilon$*, we require*

$$n = \Omega \left( \frac{1}{\delta^2} \log \left( \frac{1}{1 - \epsilon} \right) + \frac{1}{\delta} \sum_{j=1}^{L} (c_j - 1)\rho_j + \sqrt{\frac{1}{\delta^2} \log \left( \frac{1}{1 - \epsilon} \right) \cdot \frac{1}{\delta} \left( \sum_{j=1}^{L} (c_j - 1)\rho_j \right)} \right) \tag{17}$$

*number of samples for the best possible detector which is likelihood-ratio-based as mentioned in (14), for any* $\epsilon \in [0.5, 1)$*. Therefore, AI-generated text detection is possible for any* $\delta > 0$*.*

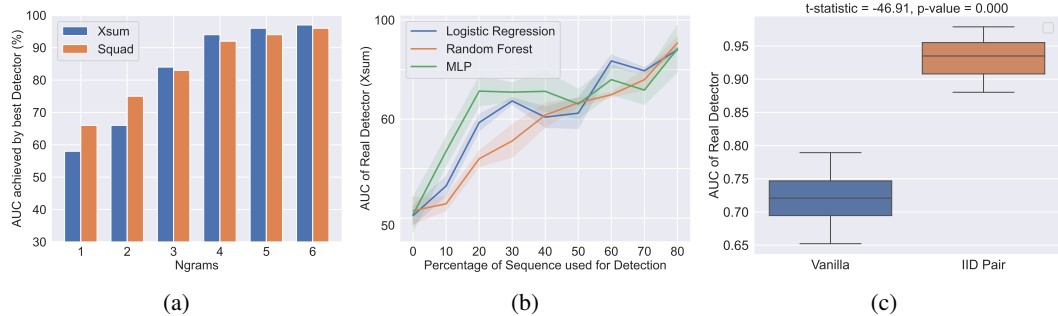

(a)  (b)  (c)

Figure 2: **(a)-(c)** validates our theorem for real human-machine classification datasets generated with XSum (Narayan et al., 2018) and Squad (Rajpurkar et al., 2016), showing that with an increase in the number of samples/sequence length, detection performance improves significantly. Figure 2a shows that the AUROC achieved by the best possible detector using the equation increases significantly from 58% to 97% with an increase in the Ngrams of the feature space for both Xsum and Squad datasets. Figure 2b demonstrates the improvement in AUROC with respect to sequence length using various real detectors/classifiers. Figure 2c shows using a box-plot-based comparison that if we consider 2 iid sequences (from either machine/human) to detect instead of one, the AUROC of the real detector improves drastically from 73% to 97%, hence validating our hypothesis.

The proof is provided in Appendix 2. From the statement of Theorem 2, we note that for $\delta > 0$ ($h(s)$ and $m(s)$ are close but not exactly the same) and $\epsilon < 1$, there exists $n$ such that we can achieve high AUROC and perform the detection. In comparison to the iid result in Theorem 1, the non-iid result in Theorem 2 has an additional term that depends on $c_j$ and $\rho_j$. Clearly, for $\rho_j = 0$, the sample complexity result in Theorem 2 boils down to the result in Theorem 1.

## 4 EXPERIMENTAL STUDIES

In this section, we provide detailed empirical evidence to support our detectability claims of this work. We consider various human-machine generated datasets and general language classification datasets.

**AUROC Discussion and Comparisons:** We first try to explain the meaning of the mathematical results we obtain via simulations. For instance, we show a pictorial representation of AUROC bound we obtained in Proposition 1 and compare it against the ROC upper bound we mentioned in (9) for different values of $n$. In Figure 1, we show that even if the original distributions of human $h(s)$ and machines $m(s)$ are close in TV norm TV = 0.1, we can increase the ROC area (and hence AUROC) via increasing the number of samples we collect $n$ to perform the detection.

### 4.1 REAL DATA EXPERIMENTS

In this section, we perform a detailed experimental analysis and ablation to validate our theorem with several real human-machine generated datasets as well as general natural language datasets.

**Datasets, AI-Text Generators and Detectors Description:** Our experimental analysis spans across 4 critical datasets, including the news articles from XSum dataset (Narayan et al., 2018), Wikipedia paragraphs from Squad dataset (Rajpurkar et al., 2016), IMDb reviews (Maas et al., 2011), and Kaggle FakeNews dataset (Lifferth, 2018), utilizing the datasets in a diverse manner to validate our hypothesis. The first two datasets (XSum and Squad) have been leveraged to generate machine-generated text by prompting an LLM with the first 50 tokens of each article in the dataset, sampling from the conditional distribution of the LLMs, as followed in (Mitchell et al., 2023; Krishna et al., 2023; Sadasivan et al., 2023). Specifically, we use a diverse set of SOTA open-source text generators including GPT-2, GPT-3.5-Turbo, Llama, Llama-2-13B-Chat-HF, and Llama-2-70B-Chat-HF as the LLM for generating the machine-generated text using the token prompts as described above. We consider 500 passages from both the Xsum and Squad datasets and subsequently 500 machine-generated texts corresponding to them using GPT-2 and evaluate the detection performance in 3

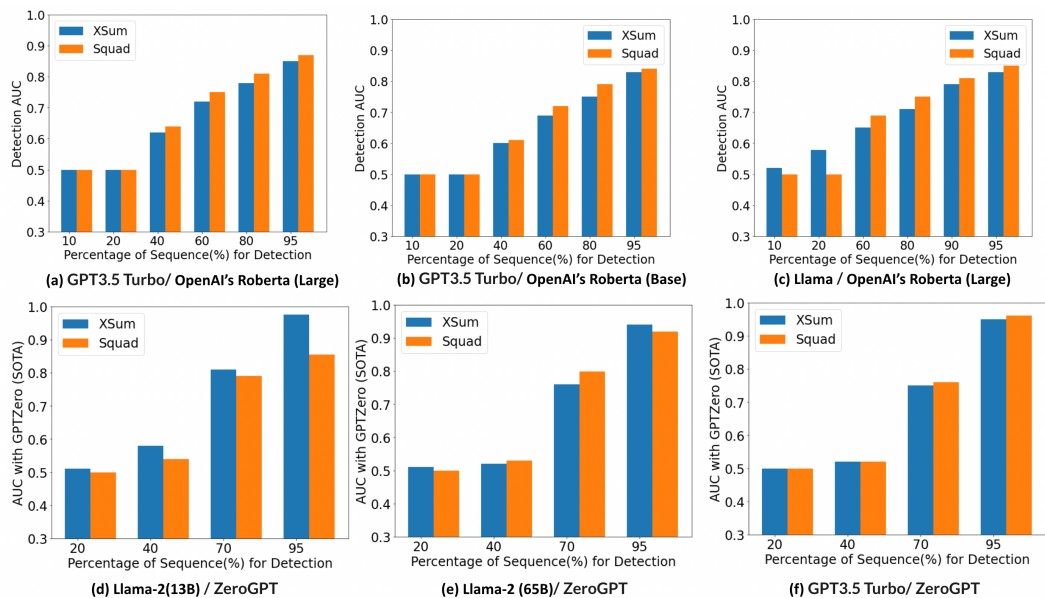

Figure 3: **(a)-(f)** validates our theorem for real human-machine classification datasets generated with XSum & Squad, with zero-shot detection performance. We use different generator/detector pairs to show the performance comparisons. For instance, (a) shows the detection performance (AUROC) of OpenAI's Roberta detector (Large) on the text generated by GPT3.5 Turbo, and we extend it to other pairs in (b)-(f). We observe that with the increase in the number of samples or sequence length for detection, the zero-shot detection performance from both the models improves from around 50% to 90% for both Xsum and Squad human-machine datasets. We also performed similar experiments with GPT-2 as well and results are available in Figure 9 in the appendix.

broad categories including *(1) supervised detection, (2) contrastive with i.i.d. samples, and (3) zero-shot performance*. Finally, we leverage two additional general language datasets (detailed in Appendix C.1), IMDb and Kaggle FakeNews, to give more insights into the separability and detection performance with an increasing number of samples.

**(1) Supervised detection performance:** To validate our hypothesis from a supervised detection/-classification perspective, we first compute the total variation distance between the human and machine-generated texts at various n-gram levels where `n-gram = 1` indicates the detection is at a word level, and as we increase it, it approaches sentence to paragraph level. We subsequently estimate the `AUROC` of the best detector using equation (14) by increasing the length of the `n-gram` from 1 to 6 as shown in Figure 2a. It is evident that with increasing `n-grams`, the `AUROC` of the best detector increases significantly from $58\%$ to $97\%$ for both Xsum and Squad datasets. This empirical observation completely aligns with our theory and intuition. To further test our hypothesis with real detectors, we train 3 vanilla classification models including Logistic Regression, Random Forest, and a 2-layer Neural Network with TF-IDF-based feature representation (bag of words) on the human-machine generated datasets including Xsum and Squad. We report the performance of the test `AUROC` with increasing sequence length in Figure 2b, which shows a significant increase in accuracy as the sequence length increases even with real detectors. This observation is also supported by the results obtained from Open-AI and summarized in the report (Solaiman et al., 2019). This impressive performance is fully aligned with our claims and provides evidence that designing a detector with high performance for AI-generated text is always possible.

**(2) Detection with pairwise IID Samples:** We also design an experiment where we assume that one can have access to 2 iid samples (from machine or human) for detection instead of just one example, which is practical and can be easily obtained in several scenarios. For example, consider detecting fake news or propaganda from a Twitter bot. We restructure our training set of the human-machine dataset by constructing pairwise training samples with labels of humans and machines and perform binary classification with only $30\%$ of the enhanced pairwise dataset with very limited bag-of-word

based features and Logistic regression, as shown in Figure 2c. We note that there is a statistically significant boost in detection performance with pairwise samples, even with a vanilla model and sampled dataset, which indicates that detection will be almost always possible in most scenarios where it is indeed crucial.

**(3) Zero-Shot detection performance:** Next, we substantiate our claims using zero-shot detection performance on the human-machine dataset for both Xsum and Squad demonstrated in Figures 3(a)-(f). For the zero-shot detection in Figures 3(a)-(c), we use the RoBERTa-Large-Detector and RoBERTa-Base-Detector from OpenAI, which are trained or fine-tuned for binary classification with datasets containing human and AI-generated texts (AIT, b). We also perform experiments with another state-of-the-art detector called ZeroGPT (AIT, a) shown in Figures 3(d)-(f). We observe that with the increase in the number of samples or sequence length of detection, the zero-shot detection performance of models improves drastically from around $50\%$ to $90\%$ on both Xsum and Squad human-machine datasets. Naturally, the performance of RoBERTa-Large-Detector is better compared to RoBERTa-Base-Detector, but still, the improvement in AUROC with the number of samples/sequence length is significant with both the models, validating our claims.

**(4) Detection with Paraphrasing:** We also perform the experiments with paraphrasing the document generated by the machine using a pre-trained Open-sourced Hugging-Face Paraphraser *Parrot* (Damodaran, 2021) which allows controlling the adequacy, fluency, and diversity of the generated text. We perform both supervised (Appendix), with pairwise IID Samples (Appendix) and Zero-shot detection with OpenAI's RoBERTa-Large-Detector. It is evident from Figure 4 that the detection performance decreases with paraphrasing as also shown in (Sadasivan et al., 2023; Krishna et al., 2023).

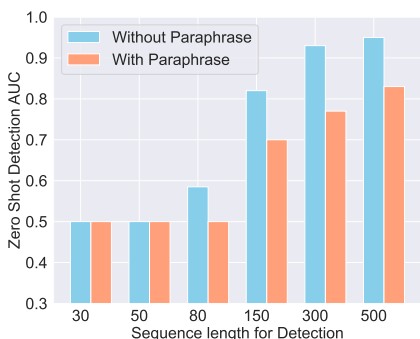

Figure 4: This figure demonstrates zero-shot detection performance with and without paraphrasing using RoBERTa-Large-Detector. Although the detection performance drops by approximately 15% due to paraphrasing, the trend of performance improvement holds as the sequence length increases.

Although the detection performance drops by approximately 15% due to paraphrasing, the trend of performance improvement still remains prominent as the sequence length increases, which validates our hypothesis even under attack. Hence, one can potentially evade such attacks by considering larger sequence lengths with the sample complexity trade-off. Additionally, we observed that the performance degradation is much lesser with pairwise iid samples, highlighting the possibilities with fine-grained detectors.

## 5 Conclusion

We note that it becomes harder to detect the AI-generated text when $m(s)$ is close to $h(s)$, and paraphrasing or successive attacks can indeed reduce the detection performance as shown in our experiments. However, in several domains where we assert that by collecting more samples/sentences it will be possible to increase the attainable area under the receiver operating characteristic curve (AUROC) sufficiently greater than $1/2$, and hence make the detection possible. We further remark that it would be quite difficult to make LLMs exactly equal to human distributions due to the vast diversity within the human population, which may require a large number of samples from an information-theoretic perspective and provide a lower bound on the closeness distance to human distributions. While there are potential risks associated with detectors, such as misidentification and false alarms, we believe that the ideal approach is to strive for more powerful, robust, fair, and better detectors and more robust watermarking techniques. To that end, we are hopeful, based on our results, that text detection is indeed possible under most of the settings and that these detectors could help mitigate the misuse of LLMs and ensure their responsible use in society.

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

# APPENDIX

## A  ADDITIONAL INSIGHTS AND REMARKS

**Remark 1: Insights for watermark design.** From a practical perspective, even though Theorem 1 shows that detection is always possible by collecting more samples, it might be costly as well if the number $n$ needed is extremely high. However, one could mitigate this trade-off by developing efficient watermarking techniques as discussed in (Kirchenbauer et al., 2023; Aaronson, 2022), which essentially increases the Chernoff information, or in other words, increases the $\delta$, eventually reducing the required number of samples. Nevertheless, empirical demonstrations in (Sadasivan et al., 2023; Krishna et al., 2023) exposed the vulnerability of the watermark-based detectors with paraphrasing-based attacks, raising a genuine concern in the community about the detection of AI-generated texts.

To address this concern, more recently, interesting work by (Krishna et al., 2023) proposed a novel defense mechanism based on information retrieval principles to combat prior attacks and demonstrated its effectiveness even with a corpus size of 15M generations. This result also supports our theory, indicating that it is always possible to detect AI-generated text depending on the detection method. In addition, there are some recent open-sourced text detection tools (AIT, a;b) whose performances are also worth considering and validate the fact that detection is indeed possible under certain settings. We believe that with the new insights from this work, one can design more efficient and robust watermarks spanning a larger corpus of text, which will be hard to remove via vanilla paraphrasers.

**Remark 2: Insights for detector design.**

This work demonstrates that detecting AI-generated text should be almost always possible but one would need to collect more samples depending on the hardness of the problem (controlled by the closeness of human and machine distributions). The recent study by (Liang et al., 2023) raises an important concern regarding the bias in some of the existing detectors. The authors in Liang et al. (2023) revealed that a significant proportion of the current detectors inaccurately classify non-native English writing samples as AI-generated, potentially leading to unjust consequences in various contexts. Interestingly, updating text generated by non-native speakers with prompts such as *Enhance it to sound more like that of a native speaker* leads to a substantial decrease in misclassification. This evidence suggests that most current detectors prioritize low perplexity as a crucial criterion for identifying a text as AI-generated, which might be flawed in various contexts, for example - academic papers as shown in (Liang et al., 2023). More specifically, we want to highlight the potential for bias in detectors relying primarily on perplexity scores, as elaborated in (Liang et al., 2023), underscoring the need for a comprehensive and equitable redesign that takes into account other relevant metrics. Our research demonstrates a promising approach to text detection, wherein the collection of more samples and the development of a multi-sample-based detector significantly enhance performance from the best word-level detector, as demonstrated by our experimental results depicted in Figures 6-8. While our results demonstrate the potential for improved detection accuracy at the paragraph level, it is important to note that this approach requires designing detectors capable of processing multiple samples. For instance, in our IMDb example, we developed a paragraph-level detector that can take the entire paragraph as input, in contrast to the word-level detector, which only processes one word at a time. Thus our approach requires the detector to deal with $n$ samples, which may be complicated compared to processing just one sample, leading to a trade-off that could be critical for accurate detection in practice. To summarize, our work offers valuable insights into detector design, specifically about the sample complexity of AI-text detention and its connection to Chernoff information of human and machine distributions. We can utilize these insights to develop robust and fair detectors that enhance the overall accuracy of text detection methods.

**Remark 3: Task-specific detectability & optimistic view of LLMs.**

In addition to our findings on the detectability of LLM-generated content, we want to highlight the significance of task-specific detectability (Figure 5). While the primary focus of Theorem 1 is to detect machine-generated text, it is important to consider the broader context of LLMs and their potential positive applications. LLMs have demonstrated significant potential to assist in a variety of tasks, including language translation (Vaswani et al., 2017), text summarization (Rush et al., 2015), dialogue systems (Serban et al., 2015), question answering (Pandya & Bhatt, 2021; Wang, 2022; Karpukhin et al., 2020), information retrieval (Chowdhury et al., 2022; Zheng et al., 2022; Kim et al.,

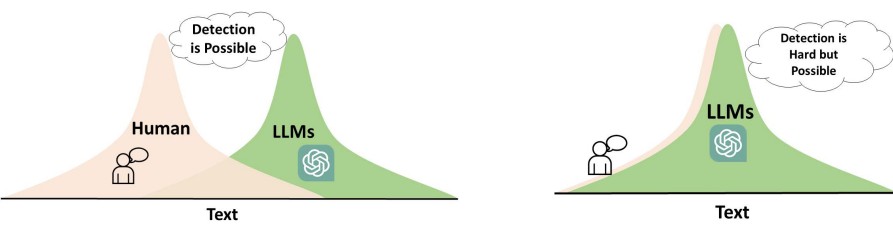

(a) LLM learns different distribution.  (b) LLM learns similar distribution.

Figure 5: We present the two detectability regimes for LLMs. Figure 5(a) denotes the scenario in which, when LLMs learn a different distribution, and the detection is easy. Figure 5(b) shows a scenario when LLMs' distribution is very close to human's, it is hard but possible to detect in this setting via collecting more samples. Additionally in scenarios of Figure 5(b), efficient watermarking techniques such as (Kirchenbauer et al., 2023; Krishna et al., 2023) could help in improving the separability and detectability.

2022), recommendation engine (Kang & McAuley, 2018; Brown et al., 2020), language grounded robotics (Ahn et al., 2022) and many others. In these scenarios, the goal is to generate high-quality text that meets the needs of the user rather than to deceive or mislead. For example, consider the application of an LLM as a tool to assist individuals or groups with moderate English writing skills to improve their writing. In this case, a well-trained LLM model could have a better (and different) distribution across $\mathcal{S}$ than the human distribution $h(s)$. This difference in distributions ensures that it should be possible to detect that AI generates the text. This understanding of detectability underscores the complexity of working with LLMs and emphasizes the importance of tailored approaches to maximize their potential. Our work provides insights into the intricacies of LLM-generated content detection, paving the way for more targeted and practical applications of these powerful models.

**Remark 4: Realistic scenarios where $m(s)$ and $h(s)$ are different.**

Theorem 1 suggests that even small differences between the machine-generated text $m(s)$ and the human-generated text $h(s)$ should help for AI-generated text detection. In many practical applications, this difference can be easily achieved since we can control $m(s)$, but not necessarily $h(s)$. One such application is the use of LLMs to address biases and prejudices in human-generated text. While biases can arise due to the diverse backgrounds of certain communities or clusters of humans, LLMs can be trained to generate unbiased text by minimizing the likelihood of biased language in the training data. This can lead to a more inclusive and equitable society, where language use is free from discrimination. Importantly, it is crucial to maintain a gap between the bias in human-generated text and that in machine-generated text. This ensures that biased language remains more likely to originate from the human-generated text than from LLMs. By doing so, effective detection and separation of the two sources can be achieved, enabling us to fully harness the potential of LLMs without compromising their integrity. With careful consideration and responsible use, LLMs can make a positive impact on our society, helping us to communicate more effectively and promoting fairness and inclusivity in language use.

## B  DETAILED PROOFS

### B.1  REVISITING LE CAM'S LEMMA AND THE EXISTENCE OF THE OPTIMAL DETECTOR

We first restate Le Cam's lemma and its proof, which appears in Le Cam (2012) and many lecture notes such as (Wasserman, 2013).

**Lemma 1** (Le Cam's Lemma). *Let $\mathcal{S}$ be an arbitrary set. For any two distributions $m$ and $h$ on $\mathcal{S}$, we have*

$$\inf_{\Psi} \left\{ \mathbb{P}_{s \sim m}[\Psi(s) \neq 1] + \mathbb{P}_{s \sim h}[\Psi(s) \neq 0] \right\} = 1 - TV(m, h), \tag{18}$$

*where the infimum is taken over all detectors (measurable maps) $\Psi : \mathcal{S} \to \{1, 0\}$. Particularly, the detector with the acceptance region $A^* := \{s : m(s) \geq h(s)\}$, defined as*

$$\Psi^*(s) := \begin{cases} 1 & s \in A^* \\ 0 & s \in \mathcal{S} \backslash A^*, \end{cases}$$

*achieves the infimum. We note that $\Psi^*$ is the likelihood ratio-based detector.*

*Proof.* For notation simplicity, we use $m$ and $h$ to denote both the probability measure and the probability density of the machine-generated and human-generated text, respectively, with the specific meaning discernible from the context. For any detector $\Psi : \mathcal{S} \to \{1, 0\}$, denote $A$ as its acceptance region, where $\Psi(s) = 1$ for $s \in A$, and $\Psi(s) = 0$ for $s \in \mathcal{S} \backslash A$. Then we have

$$\mathbb{P}_{s \sim m}[\Psi(s) \neq 1] + \mathbb{P}_{s \sim h}[\Psi(s) \neq 0] = m(\mathcal{S} \backslash A) + h(A)$$
$$= 1 - (m(A) - h(A)). \tag{19}$$

Taking the infimum over all acceptance regions on both sides in (19) yields

$$\inf_{\Psi} \left\{ \mathbb{P}_{s \sim m}[\Psi(s) \neq 1] + \mathbb{P}_{s \sim h}[\Psi(s) \neq 0] \right\} = \inf_{\Psi} \left\{ 1 - (m(A) - h(A)) \right\}$$
$$= 1 - \sup_{\Psi} \left\{ (m(A) - h(A)) \right\}$$
$$= 1 - \mathsf{TV}(m, h).$$

Next, we proceed to show that the ration-based detector $\Psi^*(s)$, defined in the statement of Lemma 1, achieves the infimum. We first note that the acceptance region $A^* := \{s : m(s) \geq h(s)\}$ is a measurable set that is included in the collection of all acceptance regions since $\mathbb{I}_{\{m(s) \geq h(s)\}}$ is a measurable function. Therefore,

$$\mathbb{P}_{s \sim m}[\Psi^*(s) \neq 1] + \mathbb{P}_{s \sim h}[\Psi^*(s) \neq 0] \geq \inf_{\Psi} \left\{ \mathbb{P}_{s \sim m}[\Psi(s) \neq 1] + \mathbb{P}_{s \sim h}[\Psi(s) \neq 0] \right\}. \tag{20}$$

On the other hand, for any measurable set $B$, we have $B \backslash A^* = \{s \in B : m(s) < h(s)\}$ and $A^* \backslash B = \{s \notin B : m(s) \geq h(s)\}$ by the definition of $A^*$. Therefore, by the sigma-additivity of measure, we have

$$1 - (m(A^*) - h(A^*)) = 1 - (m(A^* \cap B) - h(A^* \cap B)) - (m(A^* \backslash B) - h(A^* \backslash B)). \tag{21}$$

In the right-hand side of (21), we note that $m(A^* \backslash B) - h(A^* \backslash B) \geq 0$ because our detector is likelihood-ratio-based. This implies we can upper bound the right-hand side in (21) by dropping the negative term as follows

$$1 - (m(A^*) - h(A^*)) \leq 1 - (m(A^* \cap B) - h(A^* \cap B)). \tag{22}$$

Further from the definition of the ratio-based detector, we note that $m(B \backslash A^*) - h(B \backslash A^*) < 0$. This implies $-(m(B \backslash A^*) - h(B \backslash A^*)) > 0$ and we can upper bound the right hand side of (22) by adding just the positive number $-(m(B \backslash A^*) - h(B \backslash A^*))$ as follows,

$$1 - (m(A^*) - h(A^*)) \leq 1 - (m(A^* \cap B) - h(A^* \cap B)) - (m(B \backslash A^*) - h(B \backslash A^*)). \tag{23}$$

From the sigma-additivity of measure, we can write

$$1 - (m(A^*) - h(A^*)) \leq 1 - (m(B) - h(B)). \tag{24}$$

Since the inequality in (24) holds for any measurable set $B$, we can write

$$\mathbb{P}_{s \sim m}[\Psi^*(s) \neq 1] + \mathbb{P}_{s \sim h}[\Psi^*(s) \neq 0] \leq \inf_{\Psi} \left\{ \mathbb{P}_{s \sim m}[\Psi(s) \neq 1] + \mathbb{P}_{s \sim h}[\Psi(s) \neq 0] \right\}. \tag{25}$$

Hence, from the lower bound in (20) and upper bound in (25), we conclude that $\Psi^*(s)$ achieves the infimum, which completes the proof. $\qquad \square$

The Le Cam's lemma directly applies to our detector $D$ with threshold $\gamma$ by noting that any detector can be implemented via a detector with a threshold. Indeed, define $D_\gamma : \mathcal{S} \to \{1, 0\}$ via

$$D_\gamma(s) := \begin{cases} 1 & D(s) \geq \gamma \\ 0 & D(s) < \gamma, \end{cases}$$

then it holds that $\{\Psi : \mathcal{S} \to \{1,0\}\} \subseteq \{D_\gamma : \mathcal{S} \to \{1,0\}, D : \mathcal{S} \to \mathbb{R}, \gamma \in \mathbb{R}\}$ because for any $\Psi$, we can choose $D$ to be exactly the same as $\Psi$ (since $\{1,0\} \in \mathbb{R}$) and set $\gamma = 0.5$.

In fact, the detector $\Psi^*$ is exactly the likelihood-ratio-based detector which, by the Neyman-Pearson lemma (Cover, 1999, Chapter 11), is optimal in this (simple-vs.-simple) hypothesis test setting.

**Relationship to the tightness analysis in Sadasivan et al. (2023).** The authors of Sadasivan et al. (2023) provide a tightness analysis for their AUROC upper bound. The main part of the proof is to show the tightness of Equation 18. Specifically, for any given human-generated text distribution $h$, they construct a machine-generated text distribution $m$ and a detector $D$ with some threshold $\gamma$, and show that the detector with the threshold achieves the equality in Equation 18. We note that their constructed detector with the threshold is exactly the likelihood-ratio-based detector. Moreover, a key difference between our result and theirs is that we show that the tightness can be achieved for any given distribution of $m$ and $h$ while they construct a specific $m$ given $h$. While their specific construction of the machine distribution gives many insights into the problem, it is not necessary for achieving the tightness. This difference also implies that we can be more optimistic about the problem since the classifier achieving the tightness exists for any machine-generated distributions.

### B.2 PROOF OF THEOREM 1

The first part of the proof follows from the standard application of Chernoff's bounds (Vadhan, 1999, Appendix A). From the statement of Theorem 1, we note that the AUROC of the best possible detector is given by

$$\text{AUROC} = \frac{1}{2} + \text{TV}(m^{\otimes n}, h^{\otimes n}) - \frac{\text{TV}(m^{\otimes n}, h^{\otimes n})^2}{2}. \tag{26}$$

Let us start in a hard detection setting where $m(s)$ and $h(s)$ are really close and we know that $\text{TV}(m, h) = \delta$ where $\delta > 0$ is small. From the definition of TV distance, we know that there exists some set $A \in \mathcal{S}$ such that given the samples $s^m \sim m(s)$ and $s^h \sim h(s)$ it holds

$$\mathbb{P}(s^m \in A) - \mathbb{P}(s^h \in A) = \delta. \tag{27}$$

Let us define $\mathbb{P}(s^h \in A) = p$ which implies that $\mathbb{P}(s^m \in A) = p + \delta$. Let us now collect $n$ samples $\{s_i\}_{i=1}^n$ from $m(s)$, we know that the probability of any sample $s_i$ in $A$ is given by $p + \delta$. Hence, on average $(p + \delta)n$ number of samples will be in $A$. In a similar manner, if we have $n$ samples from $h(s)$, $pn$ will be in $A$ on average. Therefore, we can utilize the Chernoff bound to write

$$\mathbb{P}\left( \text{at least } \left(p + \frac{\delta}{2}\right) n \text{ samples of } h \text{ are in } A \right) \leq \exp^{\frac{-n\delta^2}{2}}$$

$$\mathbb{P}\left( \text{at most } \left(p + \frac{\delta}{2}\right) n \text{ samples of } m \text{ are in } A \right) \leq \exp^{\frac{-n\delta^2}{2}}. \tag{28}$$

Now, let us denote the set of $n-$tuples by $A'$ which contains more than $\left(p + \frac{\delta}{2}\right) n$ samples of $A$. Therefore, we can bound

$$\begin{aligned} \text{TV}(m^{\otimes n}, h^{\otimes n}) &\geq \mathbb{P}(\{s_i^m\}_{i=1}^n \in A') - \mathbb{P}(\{s_i^h\}_{i=1}^n \in A') \\ &\geq (1 - \exp^{\frac{-n\delta^2}{2}}) - \exp^{\frac{-n\delta^2}{2}} \\ &= 1 - 2\exp^{\frac{-n\delta^2}{2}}. \end{aligned} \tag{29}$$

The TV norm lower bound in (29) tells us the minimum value of $\text{TV}(m^{\otimes n}, h^{\otimes n})$ for given $n$ and $\delta$. Therefore, if we need to obtain the AUROC of the best possible detector to be equal to, or higher than say $\epsilon \in [0.5, 1]$, which means we want

$$\frac{1}{2} + \text{TV}(m^{\otimes n}, h^{\otimes n}) - \frac{\text{TV}(m^{\otimes n}, h^{\otimes n})^2}{2} \geq \epsilon. \tag{30}$$

Now, since the left-hand side is the monotonically increasing function of $\text{TV}(m^{\otimes n}, h^{\otimes n})$, it holds from the minimum value in (29) that

$$\frac{1}{2} + (1 - 2\exp^{\frac{-n\delta^2}{2}}) - \frac{(1 - 2\exp^{\frac{-n\delta^2}{2}})^2}{2} \geq \epsilon. \tag{31}$$

After expanding the squares, we get

$$\frac{1}{2} + (1 - 2\exp^{\frac{-n\delta^2}{2}}) - \frac{1}{2} - 2\exp^{-n\delta^2} + 2\exp^{\frac{-n\delta^2}{2}} \geq \epsilon. \tag{32}$$

After rearranging the terms, we get

$$\frac{1-\epsilon}{2} \geq \exp^{-n\delta^2}. \tag{33}$$

Taking log on both sides and rearranging terms yields

$$n \geq \frac{1}{\delta^2} \log\left(\frac{2}{1-\epsilon}\right). \tag{34}$$

Hence proved.

### B.3 PROOF OF THEOREM 2

Before starting the analysis, let us restate the following bound from (Dhurandhar, 2013) for quick reference.

**Lemma 2** (**Upper Bound for Non-iid scenario**). *Let $n$ be the number of samples drawn sequentially from $\mathbb{P}(S_1, S_2 \cdots S_n) = \prod_{j=1}^{L} \tau_j$, where $\tau_j$ are independent subsets consisting of $c_j$ dependent sequences $(s_1, s_2 \cdots s_{c_j})$ such that $\sum_{j=1}^{L} c_j = n$. Under dependence structure in (16), for any $\delta > \frac{\sum_{l=1}^{L}(c_j-1)\rho_j}{n}$, it holds that*

$$\mathbb{P}(|\bar{S} - \mathbb{E}[\bar{S}]| \geq \delta) \leq 2\exp\frac{-2(n\delta - \sum_{j=1}^{L}(c_j-1)\rho_j)^2}{n}, \tag{35}$$

*where $\bar{S} = \frac{1}{n}\sum_{n=1}^{n} s_i$ and $\mathbb{E}[S_i|S_{i-1} = s_{i-1}, \cdots, S_1 = s_1] = \frac{\rho}{i-1}\sum_{k=1}^{i-1} s_k + (1-\rho)\mathbb{E}[S_i]$.*

Lemma B.3 provided upper bounds for non-iid scenarios, with an exponential bound in sample size $n$ along with an additional dependence on the strength of association $\rho_j$ and the size of the dependent sequence $c_j$. It is important to note that when we have $\rho = 0$, it exactly boils down to the standard Chernoff bound.

Now, we move to do the sample complexity analysis for the non-iid setting. Similar to the proof for the iid case, we define $\mathbb{P}(s^h \in A) = p$ which implies that $\mathbb{P}(s^m \in A) = p + \delta$. Let us now collect $n$ samples sequentially $\{s_i\}_{i=1}^{n}$ from $m(s)$, we know that the probability of any sample $s_i$ in $A$ is given by $p + \delta$. Hence, on average $(p + \delta)n$ number of samples will be in $A$. In a similar manner, if we have $n$ samples from $h(s)$, $pn$ will be in $A$ on average. Therefore, we can utilize the Chernoff bound to write

$$\mathbb{P}\left(\text{at least } \left(p + \frac{\delta}{2}\right) n \text{ samples of } h \text{ are in } A\right) \leq 2e^{\frac{-2\left(n\frac{\delta}{2} - \sum_{j=1}^{L}(c_j-1)\rho_j\right)^2}{n}}$$

$$\mathbb{P}\left(\text{at most } \left(p + \frac{\delta}{2}\right) n \text{ samples of } m \text{ are in } A\right) \leq 2e^{\frac{-2\left(n\frac{\delta}{2} - \sum_{j=1}^{L}(c_j-1)\rho_j\right)^2}{n}}, \tag{36}$$

where for simplicity of notations let's consider $\beta = -\frac{2\left(n\frac{\delta}{2} - \sum_{j=1}^{L}(c_j-1)\rho_j\right)^2}{n}$. Now, let us denote the set of $n$ tuples by $A'$ which contains more than $\left(p + \frac{\delta}{2}\right) n$ samples of $A$. Therefore, we can bound

$$\begin{aligned}\mathsf{TV}(m^{\otimes n}, h^{\otimes n}) &\geq \mathbb{P}(\{s_i^m\}_{i=1}^{n} \in A') - \mathbb{P}(\{s_i^h\}_{i=1}^{n} \in A') \\ &= 1 - 4\exp^{\beta}.\end{aligned} \tag{37}$$

The TV norm lower bound in (37) tells us the minimum value of $\mathsf{TV}(m^{\otimes n}, h^{\otimes n})$ for given $n$ and $\delta$. Therefore, to obtain the AUROC of the best possible detector to be equal to, or higher than say $\epsilon \in [0.5, 1]$, it should hold that

$$\frac{1}{2} + \mathsf{TV}(m^{\otimes n}, h^{\otimes n}) - \frac{\mathsf{TV}(m^{\otimes n}, h^{\otimes n})^2}{2} \geq \epsilon. \tag{38}$$

Since the left-hand side in (38) is the monotonically increasing function of $\mathsf{TV}(m^{\otimes n}, h^{\otimes n})$, it holds from the minimum value in (29) that

$$\frac{1}{2} + (1 - 4\exp^\beta) - \frac{(1 - 4\exp^\beta)^2}{2} \geq \epsilon. \tag{39}$$

After expanding the squares, we get

$$\frac{1}{2} + 1 - 4\exp^\beta - \frac{1}{2} - 8\exp^{2\beta} + 4\exp^\beta \geq \epsilon, \tag{40}$$

which implies

$$\frac{1-\epsilon}{8} \geq \exp^{2\beta} = \exp^{-\frac{4\left(n\frac{\delta}{2} - \sum_{j=1}^L (c_j-1)\rho_j\right)^2}{n}}, \tag{41}$$

where substitute the value of $\beta$ and taking logarithm on both sides, we get

$$\log\left(\frac{8}{1-\epsilon}\right) \leq \frac{4}{n}\left(n\frac{\delta}{2} - \sum_{j=1}^L (c_j-1)\rho_j\right)^2 \tag{42}$$

$$= \frac{4}{n}\left(n\frac{\delta}{2} - \sum_{j=1}^L (c_j-1)\rho_j\right)^2$$

$$= n\delta^2 - 4\delta\left(\sum_{j=1}^L (c_j-1)\rho_j\right) + \frac{4}{n}\left(\sum_{j=1}^L (c_j-1)\rho_j\right)^2.$$

Let's denote $\alpha = \sum_{j=1}^L (c_j-1)\rho_j$ and $\gamma(\epsilon) = \log\left(\frac{8}{1-\epsilon}\right)$, for simplicity of calculations. The quadratic inequality from the above equation boils down to solving

$$\delta^2 n^2 - n(4\alpha\delta + \gamma(\epsilon)) + 4\alpha^2 \geq 0, \tag{43}$$

which is in the form of a standard quadratic equation and the corresponding solution is given by

$$n \geq \frac{\gamma(\epsilon)}{2\delta^2} + 2\frac{\alpha}{\delta} + \frac{1}{2\delta^2}\sqrt{(4\alpha\delta + \gamma(\epsilon))^2 - 16\alpha^2\delta^2} \tag{44}$$

$$= \frac{\gamma(\epsilon)}{2\delta^2} + 2\frac{\alpha}{\delta} + \frac{1}{2\delta^2}\sqrt{\gamma(\epsilon)^2 + 8\alpha\delta\gamma(\epsilon)}$$

$$= \frac{\gamma(\epsilon)}{2\delta^2} + \frac{2}{\delta}\sum_{j=1}^L (c_j-1)\rho_j + \frac{1}{2\delta^2}\sqrt{(\gamma(\epsilon))^2 + 8\left(\sum_{j=1}^L (c_j-1)\rho_j\right)\delta\gamma(\epsilon)}.$$

Now, we further expand upon the expression as

$$n \geq \frac{1}{2\delta^2}\gamma(\epsilon) + \frac{2}{\delta}\sum_{j=1}^L (c_j-1)\rho_j + \frac{1}{\sqrt{2}\delta^2}\sqrt{\frac{1}{2}\left((\gamma(\epsilon))^2 + 8\left(\sum_{j=1}^L (c_j-1)\rho_j\right)\delta\gamma(\epsilon)\right)}$$

$$\geq \frac{1}{2\delta^2}\gamma(\epsilon) + \frac{2}{\delta}\sum_{j=1}^L (c_j-1)\rho_j + \frac{1}{2\sqrt{2}\delta^2}\gamma(\epsilon) + \frac{1}{\sqrt{2}\delta^2}\sqrt{2\left(\sum_{j=1}^L (c_j-1)\rho_j\right)\delta\gamma(\epsilon)}, \tag{45}$$

where, the first-term results from multiplying and dividing by a constant factor 2, and the second term is an application of Jensen's inequality for convex functions. Using the order notation, we obtain

$$n = \Omega\left(\frac{1}{\delta^2}\log\left(\frac{1}{1-\epsilon}\right) + \frac{1}{\delta}\sum_{j=1}^L (c_j-1)\rho_j + \sqrt{\frac{1}{\delta^3}\log\left(\frac{1}{1-\epsilon}\right)\left(\sum_{j=1}^L (c_j-1)\rho_j\right)}\right). \tag{46}$$

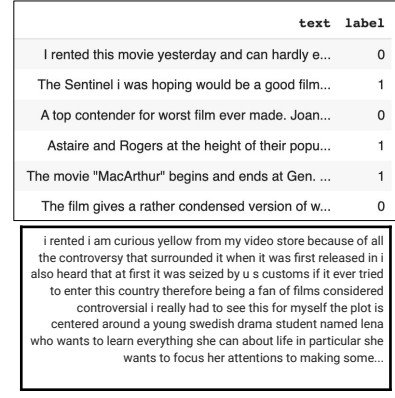
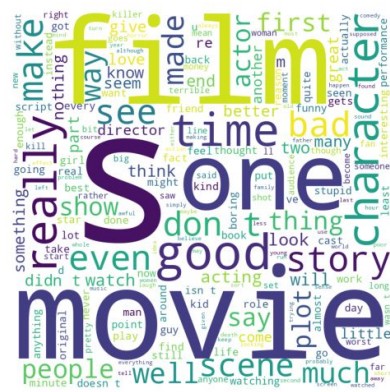

(a) IMDb dataset (Maas et al., 2011)

| able | about | absolutely | acting | action | actor | actors | actress | actually | after | ... |
|------|-------|------------|--------|--------|-------|--------|---------|----------|-------|-----|
| 0 | 0 | 1 | 0 | 0 | 0 | 0 | 0 | 0 | 0 | ... |
| 0 | 1 | 0 | 0 | 0 | 0 | 1 | 0 | 0 | 0 | ... |
| 0 | 0 | 0 | 0 | 0 | 0 | 0 | 0 | 1 | 3 | ... |
| 0 | 1 | 1 | 0 | 0 | 0 | 0 | 0 | 0 | 0 | ... |
| 0 | 1 | 0 | 0 | 0 | 0 | 1 | 0 | 0 | 1 | ... |
| 0 | 1 | 0 | 0 | 0 | 0 | 0 | 0 | 2 | 0 | ... |
| 0 | 1 | 0 | 0 | 0 | 0 | 0 | 0 | 0 | 0 | ... |
| 0 | 1 | 0 | 1 | 0 | 0 | 0 | 0 | 0 | 2 | ... |
| 0 | 0 | 0 | 0 | 0 | 0 | 0 | 0 | 1 | 0 | ... |
| 0 | 1 | 0 | 0 | 0 | 0 | 1 | 0 | 0 | 1 | ... |

(b) Paragraph representation space

Figure 6: Figure 6(a) (left) shows examples of textual paragraphs and corresponding labels present in the IMDb dataset. It also highlights part of one random paragraph (one input) showing that in general, having a lot of sentences as input for detection is very common and practical. Figure 4(a) (right) represents the word-cloud representation of the word distribution based on which the word-level total variation is estimated. Figure 6(b) denotes the representation of the input paragraph using a Bag-of-Words-based count vectorizer for our algorithms and proving our hypothesis. It demonstrates paragraph representation space with Bag-of-Words-based count vectorizer where each row indicates one review.

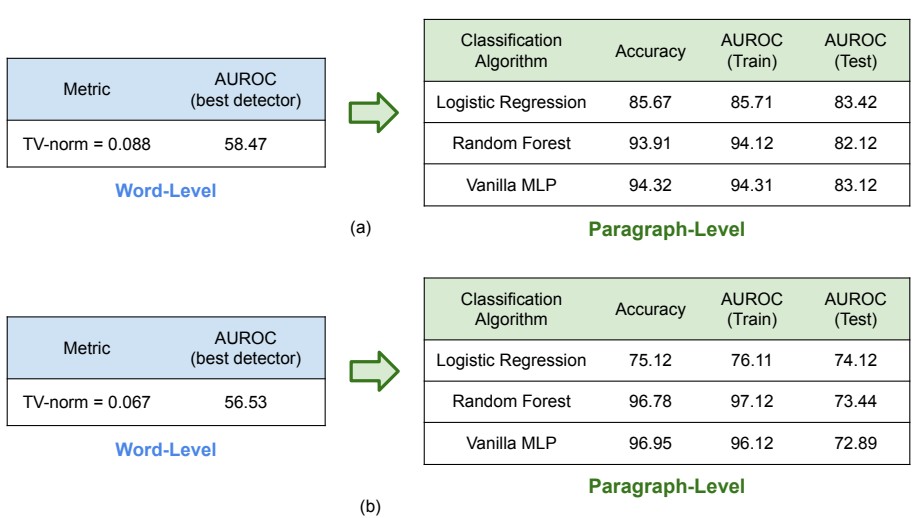

Figure 7: The table in Figure 7(a) (left) represents the total variation norm distance at a word level i.e input to the detector is the word and one needs to detect if it's a positive or negative class (human or machine in our context). It also shows the AUROC that can be achieved by the best detector based on the total variation norm as shown in (Sadasivan et al., 2023). Figure 7(b) (right) shows the accuracy and AUROC achieved by real detectors (standard machine learning algorithms) at a paragraph level, where each input to the detector is a paragraph or a group of sentences. It is evident that at a paragraph level, even a simple untuned ML detector can achieve a very high AUROC of more than 85%, which was very low at a word level. Similarly, in the tables in Figure 7(b), we observe a similar behavior as we increase the hardness of the problem by reducing the number of sentences from the passage. We note that the AUROC achieved by the real detector decreases but is still much larger than the word-level best detector's AUROC which validates our claims.

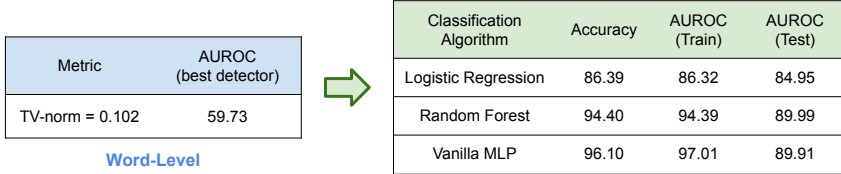

Figure 8: The table in Figure 8(a) (left) represents the total variation norm distance at a word level for the Fake News dataset (Lifferth, 2018). It shows the AUROC that can be achieved by the best detector based on the total variation norm as shown is 59.73%. Figure 8(b) (right) shows the accuracy and AUC achieved by a real detector at a paragraph level goes up to 90%, which validates our hypothesis for a general class of NLP tasks

## C  ADDITIONAL FIGURES OF EXPERIMENTAL RESULTS

### C.1  ADDITIONAL EXPERIMENTAL DETAILS

**IMDb Dataset Experiments.** To validate our claims on the possibilities of detection, we run experiments on the IMDb dataset (Maas et al., 2011), which is a widely-used benchmark dataset in the field of natural language processing. The dataset consists of $50,000$ movie reviews from the internet movie database that have been labeled as positive or negative based on their sentiment. The goal is to classify the reviews accordingly based on their text content. The experiments are done to validate our hypothesis on a more general class of language tasks including classification and detection. We specifically focus on the representation space of the inputs for both the human and machine distributions and try to validate our hypothesis by comparing the input space of words to the input space of a group of sentences. The objective is to analyze the variations in performance of the detector when detecting at word-level versus paragraphs. Hence, there are two scenarios to consider. The first is where we're given a word and we have to determine whether it came from positive or negative class. The second, and more practical case, is where we're given a paragraph i.e a group of sentences and we have to detect whether it came from positive or negative class.

So, we first compute the total variation distance between the positive and negative classes at the word level. This is done by computing the divergence between the distribution over the space of words between the two classes. Figure 7(a) shows that the best possible AUROC achieved by the detector is $0.585$ at the word level. From these results, it seems almost impossible to distinguish the two classes. However when we perform the detection at a paragraph level using a real detector (standard ML models, including random forest, logistic regression, and a vanilla multi-layer perception), we see a remarkable improvement in the detection performance. As shown in Figure 7(a), all the real detectors achieve a train AUROC of greater than $0.85$ ($\geq 0.93$ for random forest and MLP), and a test AUROC of greater than $0.8$, which surpasses the upper-bounds of the best detector at a word level, validating our theory and intuition. This impressive performance is fully aligned with our claims and provides evidence that designing a detector with high performance for AI-generated text is always possible even for general NLP detection tasks.

**IMDb NLP Dataset Experiments with Increased Hardness.** To provide additional confirmation of the efficacy of our claim, we made the experimental setting more challenging by randomly decreasing the number of sentences in each review, making it difficult for any genuine detector or classifier to distinguish. In this scenario, we again compared the performance to the previous scenario and observed that all the methods were able to achieve a test AUROC greater than 0.7, which is lower than the previous case. This result supports our hypothesis that as the number of samples/sentences increases, detection accuracy improves.

We conducted a similar experiment on a Fake News classification dataset (Lifferth, 2018), and the results were consistent with our previous findings. This indicates that AI-generated text can be detected, although we need to be cautious and gather more samples as the distribution becomes closer.

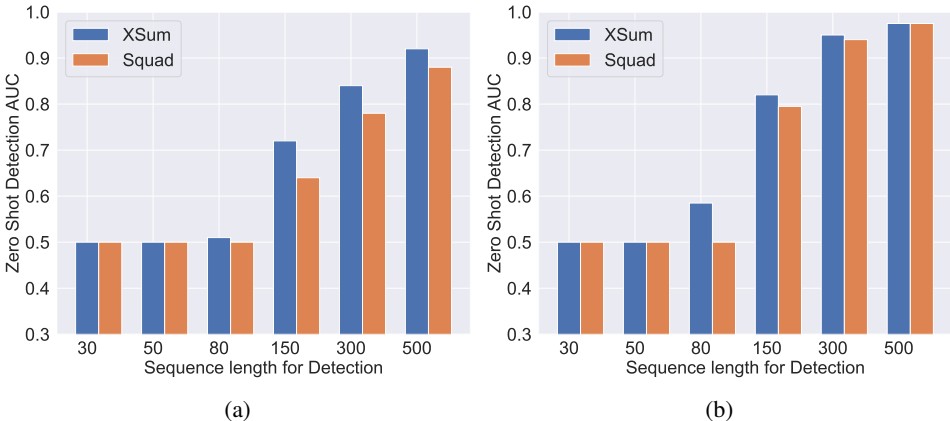

Figure 9: **(a)-(b)** validates our theorem for real human-machine classification datasets generated with XSum & Squad, with zero-shot detection performance. We use the RoBERTa-Base-Detector (9a) and RoBERTa-Large-Detector (9b) from OpenAI which are trained or fine-tuned for binary classification with datasets containing human and AI-generated texts. We observe that with the increase in the number of samples or sequence length for detection, the zero-shot detection performance from both the models improves drastically from around 50% to 97% for both Xsum and Squad human-machine datasets.

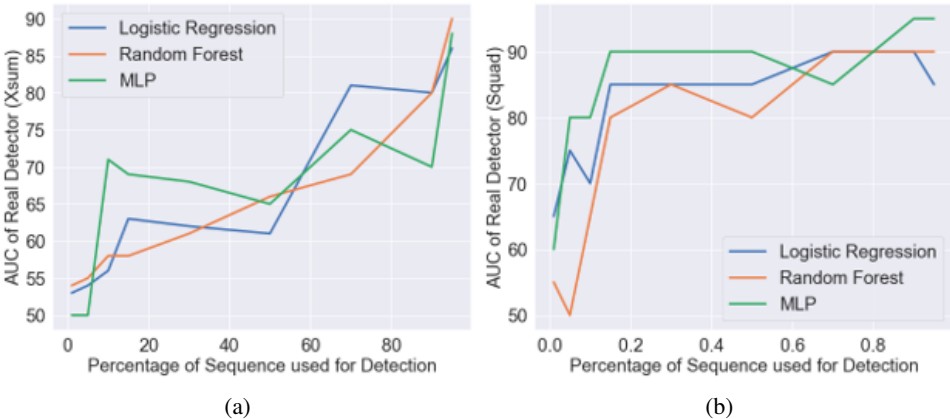

Figure 10: (a) demonstrates the detection performance of Vanilla classifiers/detectors on the Xsum dataset (Randomly sampled) generated by GPT-2. (b) demonstrates the detection performance of Vanilla classifiers/detectors on the Squad dataset (Randomly sampled) generated by GPT-2. This shows that even for vanilla detectors, our result holds for random subsets of the data.

We would like to emphasize that the purpose of this experimentation is to demonstrate our hypothesis regarding the feasibility of detection rather than to showcase the accuracy of classification. This is because the accuracy of classification is already well-established, with a simple pre-trained BERT-based model being capable of achieving high accuracy.

## D   DETAILED CONCLUSION & SCOPE OF FUTURE WORKS

We note that it becomes harder to detect the AI-generated text when $m(s)$ is close to $h(s)$, and paraphrasing attacks can indeed reduce the detection performance as shown in our experiments. However, we assert that by collecting more samples/sentences, it will be possible to increase the attainable area under the receiver operating characteristic curve (AUROC) sufficiently greater than $1/2$, and hence make the detection possible. We further remark that it would be quite difficult to make LLMs exactly equal to human distributions due to the vast diversity within the human population,

which may require a large number of samples from an information-theoretic perspective and provides a lower bound on the closeness distance to human distributions. Diversity could lead to realistic analysis to prove that the distributions are sufficiently separated to be detectable.

We want to emphasize that as we show detectability is always possible (unless $m = h$ in exactness), in several scenarios when $m$ and $h$ are very close, it might need a lot of samples to detect. However, watermark-based techniques can help address this issue by causing shifts in the distributions. The additional insights from our work could help to design better watermarks, which cannot be attacked easily with paraphrases. More specifically, it is possible to create more powerful and robust watermarks to introduce a minor change in the machine distributions, and then collecting more samples should help to perform the AI-generated text detection.

While there are potential risks associated with detectors, such as misidentification and false alarms, we believe that the ideal approach is to strive for more powerful, robust, fair, and better detectors and more robust watermarking techniques. We believe that addressing issues such as representation space, robust watermarks, and interpretability is crucial for the safe and trustworthy application of generative language models and detection. To that end, we are hopeful, based on our results, that text detection is indeed possible under most of the settings and that these detectors could help mitigate the misuse of LLMs and ensure their responsible use in society.

## E  NEW EXPERIMENTAL RESULTS

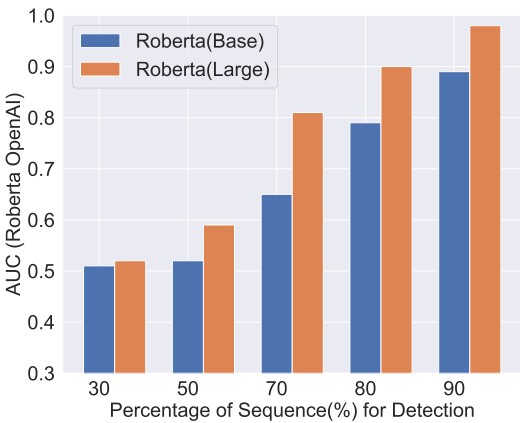

Figure 11: This figure validates our theorem for wuestion answering datasets generated with Squad, with zero-shot detection performance. We provide the question as input and generate the context from GPT-2 model and consider the context present in SQuad as human-generated to study the detection performance. We use the RoBERTa-Base-Detector (9a) and RoBERTa-Large-Detector (9b) from OpenAI which are trained or fine-tuned for binary classification with datasets containing human and AI-generated texts. We observe that with the increase in sequence length for detection, the zero-shot detection performance from both the models improves significantly from around 55% to 98% validating our theorem for QA tasks

