# OpenReview forum: "On the Possibilities of AI-Generated Text Detection: A Sample Complexity Analysis"
_ICLR.cc/2024/Conference — Submitted to ICLR 2024_

### Official Review · Reviewer_6pNs · 2023-10-15

**Soundness:** 4 excellent
**Presentation:** 4 excellent
**Contribution:** 3 good
**Rating:** 6
**Confidence:** 3

**Summary:**

This paper presents a theoretical and empirical analysis of the feasibility of AI-generated text detection as LLMs become more powerful. Overall, the paper shows that even with the most powerful LLMs, AI-generated text detection will always feasible with enough samples / long-enough generated sequences.

As some background, Sadasivan et al. 2023 [1] recently showed in their work that AI-generated text detection performance is fundamentally bounded by the total variation norm between human-written and machine-generated text. Sadasivan et al. 2023 hypothesize that as LLMs become stronger, this variation will reduce, eventually making AI-generated text impossible.

This paper builds on the theoretical analysis of Sadasivan et al. 2023, but instead includes the *number of available AI-generated / human-written samples* into their proof. The paper argues that in many real-life scenarios (like AI Twitter bot detection), it is always possible to access multiple samples of AI-generated text. The paper finds that the bound on AI-generated text detection performance exponentially increases with the number of samples. The paper also provides an equation to obtain the required number of samples to achieve a particular AU-ROC score.

The paper further supports the theoretical results with several empirical experiments, showing that AI-generated text can be better detected with more samples or with longer sequences. The experiments include 4 datasets and evaluate 5 different LLMs with classification-based detection methods.

[1] - https://arxiv.org/abs/2303.11156

**Strengths:**

1. Given the growing risk of plagiarism in college essays and the spread of misinformation using large language models, the topic of the paper is timely and very important.

2. The paper presents an optimistic theoretical take on the feasibility AI-generated text detection, and in a sense, their proof conflicts with the impossibility result of Sadasivan et al. 2023. This proof is a good motivation for researchers to continue working on AI-generated text detection, despite the fear of LLMs approach human performance.

3. The paper provides the exact theoretical bounds on AU-ROC performance, measures its tightness, and provides an equation to derive the required samples for a desired AU-ROC performance (however, also see weakness #1 on how this could be more practical).

4. The paper supports their theoretical analysis with comprehensive experimental results, showcasing the benefit of longer sequences and more samples in 4 datasets and with 5 large language models.

5. The paper is well written and a fair amount of intuition is added before/after the theoretical analysis to make it more accessible to a general audience.

**Weaknesses:**

Overall, I liked the sentiment in the paper and thought the theoretical and empirical analysis was thorough. While I am leaning accept, I had some concerns about the real-world takeaways from the work, and would love to hear the author's thoughts on it (please let me know if this was mentioned somewhere in the paper or appendix and I missed it!).

1. The theoretical analysis in the paper has a nice overall takeaway, but I am not sure how helpful the bounds are for practioners who are developing these algorithms / trying to detect AI-generated text. Specifically,

* How does one go about computing the variation between machine-generated text and human-written text (`TV(m,h)^n`) for some SoTA LLMs like ChatGPT or GPT4, perhaps in a restricted domain? This is needed in both proposition 1 (for AU-ROC upperbound calculation), and in Theorem 1 (for sample estimation). I wonder if the authors could walkthrough an example in their paper: calculate `TV(m,h)^n` for ChatGPT for different output lengths, and then provide the corresponding upperbounds for AU-ROC for different values of the output length. I wonder if the method used in the MAUVE text evaluation paper (https://arxiv.org/abs/2102.01454) may be helpful for this calculation, who propose a clever approximation for measuring the divergence between human/machine-written text.

* Is it possible to derive a bound on the true-positive rate (TPR) for a fixed false-positive rate (FPR), instead of a bound on AU-ROC? This may be a more practical metric, since practioners will want to optimize TPR for a fixed low FPR.

2. I feel like the authors are under-selling themselves in the title/abstract/introduction, by focusing on the "number of samples" rather than "sequence length". Overall, I think in many AI-generated text detection scenarios only 1 sample will be available (like essay plagiarism detection, fake news detection). However, these individual samples are likely to have multiple sentences (or even paragraphs) in them. Moreover, the general empirical consensus seems to be that longer AI-generated sequences are easier to detect (https://arxiv.org/pdf/2306.04634.pdf). Does a multi-sentence output result in `n > 1` in the author's theoretical analysis? I am suspecting this is true, since a number of empirical experiments iterate on the output length rather than the number of samples. If feasible, I recommend the authors to make output length the first-class citizen of the work, rather than number of samples.

**Questions:**

I was wondering how the retrieval-based defense from Krishna et al. 2023 (https://arxiv.org/abs/2303.13408) fits in the theoretical analysis of this paper. While I appreciate the discussion in Appendix A, I think the analysis in both Sadasivan et al. 2023 and this paper does not directly apply to retrieval-based detection. If I understand the retrieval-based detection algorithm correctly, it does not leverage the difference in properties between `m(s)` and `h(s)`. In other words, I think retrieval-based detection will work with `h_1(s)` and `h_2(s)` as well, where `h_1` and `h_2` are two identically distributed human writers. Is it fair to say that in retrieval-based detection, the support (`S` in this paper's 3.1) of `m(s)` and `h(s)` is almost disjoint, and hence the `TV(m,h)` will be quite high / infinity? With a bigger database there may be a few collisions between `m(s)` and `h(s)`, so `TV(m,h)` will slightly drop?

---

> ### Author Response · Authors · 2023-11-17
> **Response to Reviewer 6pNs (Part 1)**
>
> **General Response** We sincerely appreciate the reviewer's thoughtful feedback on our paper. We would like to acknowledge the reviewer for acknowledging the major motivation, theoretical contribution, and empirical analysis of our work, especially emphasizing that our results can motivate researchers to continue working on AI-generated text detection, despite the fear of LLMs approaching human performance --> this was our exact motivation.
>
> **Important Highlights:** We want to specifically highlight the reviewer's extremely important comment "authors are under-selling themselves in the title/abstract/introduction focusing on the "number of samples" rather than "sequence length". We absolutely agree that our non-iid results are much more general and generalize to relating the sample complexity with sequence length and not just more samples (Th 1) which is practical and more general. We will be updating our draft with special emphasis on this point as we believe this contribution was missed by some of the reviewers.
>
> ### Weakness
>
> > Comment: Overall, I liked the sentiment in the paper and thought the theoretical and empirical analysis was thorough. While I am leaning accept, I had some concerns about the real-world takeaways from the work, and would love to hear the author's thoughts on it (please let me know if this was mentioned somewhere in the paper or appendix and I missed it!).
>
> **Response to Comment** Thanks a lot for the comment. We provide a detailed discussion below.
>
> > Weakness 1 : The theoretical analysis in the paper has a nice overall takeaway, but I am not sure how helpful the bounds are for practitioners who are developing these algorithms / trying to detect AI-generated text. We try to discuss some implications of our
>
> **Response to Weakness 1** Thanks for this comment and we agree that it is indeed very important to discuss the significance and practical implications of our result. We attempt to provide certain insights of our results in Appendix A (remarks 1, 2, 3). Here we highlight how practitioners can leverage our results in designing reliable and more robust watermarks & detectors (currently susceptible to attacks) with information gain and human-machine distribution distance, and also discuss on the task-specific detectability with our results. We give two examples for simplicity
>
> Watermark Design: Our analysis suggests adaptive watermark designs based on Chernoff information and sample complexity, enhancing robustness against attacks—a future research avenue.
>
> Length Constraints in Exams: Our bounds inform optimal summary lengths for easier cheating detection, leveraging the Chernoff metric.
>
> However, we agree that a more specific and detailed discussion on the practicality will be critical and we will add the same in our updated draft.
>
> > Weakness 2: How does one go about computing the variation between machine-generated text and human-written text (TV(m,h)^n) for some SoTA LLMs like ChatGPT or GPT4, perhaps in..... corresponding upper bounds for AU-ROC for different values of the output length. I wonder if the method used in the MAUVE text evaluation paper (https://arxiv.org/abs/2102.01454) may be helpful for this calculation, who proposes a clever approximation for measuring the divergence between human/machine-written text.
>
> **Response to Weakness 2** This is a very interesting point. We agree that computing the total variation distance in exactness for a real detector for text generated from LLMs or humans is hard and hence we evaluate the performance w.r.t AUC similar to what was used in prior works on detection (Sadasivan et. al, Krishna et. al, Mitchell et al.).
>
> However, it is important to note that we do validate the AUC upper-bound (in exactness, missing from concurrent works) for the best detector with an n-gram bag of words based on discretized feature space in Figure 2a. Since, in a finite discretized feature space, TV distance can be computed using the definition [1] and then we plot the AUC upper bounds by varying the value of n in n-grams as in Figure 2a.
>
> On the other hand, the MAUVE metric leverages a modified KL/JS type divergence as in Equation 2 (Pillutla et al 2021), which can be an alternate measure of divergence replacing TV distance. Thanks for this very interesting suggestion and we agree that due to the piecewise constant approximation of the human (machine) distributions as done in Equation 3 (Mauve paper), we can estimate the KL divergence (or TV distance) under some approximations (depending on the clustering), and is a valid scope of future research. Additionally, one can also show theoretically by replacing TV with KL using Pinsker's inequality and obtaining the equivalent upper bound like ours in Theorem 1,2.

---

> > ### Author Response · Authors · 2023-11-17
> > **Response to Reviewer 6pNs (Part 2)**
> >
> > > Weakness 3: Is it possible to derive a bound on the true-positive rate (TPR) for a fixed false-positive rate (FPR), instead of a bound on AU-ROC? This may be a more practical metric, since practioners will want to optimize TPR for a fixed low FPR.
> >
> > **Response to Weakness 3** This is a very good question. We want to highlight that we already have the implication of the bound on TPR for fixed FPR in our theoretical analysis in an implicit manner in equation 9. Let us try to explain the same with specific context here: In equation 9, if we set the value of FPR to be very low let's say 0.01 (as an example), equation 9 boils down to $TPR <= min (0.01 + TV(m^n,h^n), 1)$  and naturally let's focus on the setting when TV(m,h) <1, then the bound eventually boils down to $TPR <= TV(m^n,h^n)$ i.e depends on the total variation divergence b/w the product distribution of machine and human respectively. Interestingly in equation 12, we show that with an increasing number of samples, the total variation distance between $TV(m^n,h^n)$ approaches 1 exponentially, and hence the TPR will reach close to 1 (~1 - FPR), with the same rate.
> >
> > We agree that in many of the practical scenarios, TPR for low FPR is an optimal metric to consider by practitioners and we will definitely add the discussion in our updated draft.
> >
> > > Weakness 4: I feel like the authors are under-selling themselves in the title/abstract/introduction, by focusing on the "number of samples" rather than "sequence length". Overall, I think in many AI-generated text detection scenarios only 1 sample will be available (like essay plagiarism detection, or fake news detection). However, these individual samples are likely to have multiple sentences (or even paragraphs) in them. Moreover, the general empirical consensus seems to be that longer AI-generated sequences are easier to detect (https://arxiv.org/pdf/2306.04634.pdf). Does a multi-sentence output result in n > 1 in the author's theoretical analysis? I am suspecting this is true, since a number of empirical experiments iterate on the output length rather than the number of samples. If feasible, I recommend the authors to make output length the first-class citizen of the work, rather than number of samples.
> >
> > **Response to Weakness 4** This is an extremely critical and important point and we want to thank the reviewer for highlighting this specifically. Indeed our theoretical and empirical results are much more general and not just restricted to the setting requiring iid samples. Our Theorem 2 extends to the more general non-iid setting where we relate the sample complexity with the sequence length. As the reviewer correctly pointed out, the majority of our experimental results (Figure 3a-f, 4, etc) are indeed for the non-iid settings and validate our theoretical findings. We absolutely agree with the reviewer that we need to emphasize more on Theorem 2 i.e. sample complexity with sequence length and will update our draft accordingly.
> >
> > ### Questions
> >
> > > Question 1 : I was wondering how the retrieval-based defense from Krishna et al. 2023 (https://arxiv.org/abs/2303.13408) fits in the theoretical analysis of this paper. While I appreciate the discussion in Appendix A, I think the analysis in both Sadasivan et al. 2023 and this paper does not directly apply to retrieval-based detection. If I understand the retrieval-based detection algorithm correctly, it does not leverage the difference in properties between m(s) and h(s). In other words, I think retrieval-based detection will work with h_1(s) and h_2(s) as well, where h_1 and h_2 are two identically distributed human writers. Is it fair to say that in retrieval-based detection, the support (S in this paper's 3.1) of m(s) and h(s) is almost disjoint, and hence the TV(m,h) will be quite high / infinity? With a bigger database there may be a few collisions between m(s) and h(s), so TV(m,h) will slightly drop?
> >
> >
> > **Response to Question** This is an excellent point and we agree that retrieval-based detection posits an alternative interesting way of formulating the detection problem. However, as the reviewer correctly pointed out the way retrieval has been formulated, the TV distance b/w m(s) and h(s) will be already very high (almost near 1 without perturbation/paraphrasing) due to the disjoint support leading to high AUC from equation 1. However, a rigorous theoretical analysis of the retrieval framework as a defense is an interesting and valid scope of future research.

---

> > > ### Author Response · Authors · 2023-11-19
> > > **Request to the Reviewer**
> > >
> > > Dear Reviewer,
> > >
> > > Thank you so much for your time and efforts in reviewing our paper. We have addressed your comments in detail and are happy to discuss more if there are any additional concerns. We are looking forward to your feedback and would greatly appreciate you consider raising the scores.
> > >
> > > Thank you,
> > >
> > > Authors

---

> > > > ### Author Response · Authors · 2023-11-22
> > > > **Request to the Reviewer**
> > > >
> > > > Dear Reviewer,
> > > >
> > > > Since the deadline for the author-reviewer discussion period is approaching (today is the deadline), we wanted to humbly check with the reviewer for any remaining concerns. We are happy to discuss more. We are looking forward to your feedback.
> > > >
> > > > Thank you,
> > > >
> > > > Authors

---

> > > > > ### Comment · Reviewer_6pNs · 2023-11-22
> > > > > **Thank you for your detailed response**
> > > > >
> > > > > Dear authors,
> > > > > Thank you for your detailed response to my comments. After carefully reviewing them, I don't think I'll be able to raise my score to the next highest rating of 8, since my concerns on the real-world implications of the bounds remain.
> > > > >
> > > > > My assessment is more like a 6.5-7, and I continue to support the acceptance of the paper. I will look forward to the changes in the next version of the paper!

---

> > > > > > ### Author Response · Authors · 2023-11-22
> > > > > > **Real-world Implications of the Bounds**
> > > > > >
> > > > > > >Dear authors, Thank you for your detailed response to my comments. After carefully reviewing them, I don't think I'll be able to raise my score to the next highest rating of 8, since my concerns on the real-world implications of the bounds remain. My assessment is more like a 6.5-7, and I continue to support the acceptance of the paper. I will look forward to the changes in the next version of the paper!
> > > > > >
> > > > > >
> > > > > > **Response:** Thank you so much for your thoughtful comments and for supporting the acceptance of our paper.  We appreciate this opportunity to clarify further and apologize if our earlier response was not detailed enough to clarify the concerns. We agree that this is an important point and we highlight the **three major real-world implications** of our results as follows.
> > > > > >
> > > > > > **(1)** ***Takeaways for the Watermarking Design (prepared detection)*** for AI-generated text detection: We note that watermarking (Kirchenbauer et al., 2023; Aaronson, 2022), the output of language models provides a reliable solution to the problem of AI-generated text detection. Specifically, watermarking-based methods alter the output distribution of the text (for a given prompt) generated by the language model in some predefined manner (such as dividing the vocabulary into green and red list as in Kirchenbauer et al., 2023) with the potential drawback of text quality degradation. Hence, one critical question here is, what metric or objective a watermarking scheme should utilize so that the generated text quality after watermarking remains the same is where our results provide insights.***  From the statement in Theorem 1, and 2 on Page 6 (and also from the equality in 12), we note that a watermarking scheme should focus on increasing the Chernoff Information in order to increase the probability of detection with smaller sequence length $n$.
> > > > > >
> > > > > > **(2)** ***Takeaways for the Detector Design (post hoc detection)*** for AI-generated text detection: There is another class of detection methods that try to directly classify the given text as either AI-generated or not based on some pre-trained/fine-tuned classifiers (GPTZero, OpenAI’s Roberta detector, etc.). **In this scenario, our results directly imply that** the detector would be more successful if the sequence length is higher. This essentially means that we can utilize our bounds' information to provide an estimate of **minimum possible length** required for reliably detecting the text, which would help in infusing confidence among the community and reduce the false alarms.
> > > > > >
> > > > > > **(3)** ***Our Result Advocates the Design of Task-Specific Detectors:***  A critical implication of our results (Theorem 1 and 2 on Page 6) is that it connects the performance of detectors with sequence length. Interestingly, this connection could lead to a segregation of the application domains in real world where AI-generated text detection can be easy vs hard, based on the ***sequence length and Chernoff information***. This indirectly highlights the need to study the problem of detection in a domain-specific manner and not independent of the domain. For example, in scenarios, where we can have documents of larger lengths (such as exams, news articles, conference proceedings, etc.) or multiple samples (Twitter bot, fake user reviews, etc.) it would be easier to detect than other domains where it is hard to obtain large sequence length text (such as medical diagnosis, hateful messages on social media, etc.) which requires dedicated research efforts.
> > > > > >
> > > > > > We attempted to provide a glimpse of these takeaways in our Remarks in Section  [Appendix A (remark 1, 2, 3)](https://openreview.net/pdf?id=oxEER3kZ9M). We will specifically highlight them in the main body of our final version as suggested by the Reviewer. Thank you once again for your engagement during the rebuttal period.

---

### Official Review · Reviewer_JMFi · 2023-10-31

**Soundness:** 3 good
**Presentation:** 3 good
**Contribution:** 1 poor
**Rating:** 3
**Confidence:** 3

**Summary:**

This paper provides a theoretical analysis of the feasibility of AI-generated text detection based on sample complexity and TV distance. The paper includes two main theorems, one on the possibility of sample complexity in general (which I believe is an existing known result) and the second which extends to non-IID cases. The paper also includes a few experiments on the effect of document length on detection accuracy.

**Strengths:**

The paper is clear and well-written and provides a clear justification for a known result, i.e., that AI-generated text detection is easier with longer documents. The argument that Sadasivan, et al. 2023 (“Can AI-generated Text Be Reliably Detected?”) ignores the effect of document length and the possibility of having multiple sampled documents is reasonable and well-taken.

**Weaknesses:**

My main concern is that the results in this paper are general facts about sample complexity and not specific to detection of AI-generated text. For example, Theorem 1 is included here: https://github.com/ccanonne/probabilitydistributiontoolbox/blob/master/testing.pdf. As this paper is outside of my area, I am less confident about the novelty of Theorem 2, but its formulation is not dependent on the domain of AI-generated text detection. It is also unclear from the paper whether the “multiple samples” required by this theorem are meant to be individual words, sentences, or documents.

The experiments also replicate known results, namely, that AI-generated text detectors perform better on longer sequences and that they perform worse with paraphrasing. As far as I can tell, the main novel finding is the “pairwise with IID samples” condition (Figure 2c). It would be interesting to see a more controlled experiment here, which compares the performance of detectors on two IID documents of lengths X,Y versus one single document of length X+Y.

**Questions:**

1. Could you provide additional details on the process used to generate Figure 2a? How were documents tokenized and how was AUC computed? I am also confused by the statement in Section 4.1 that this analysis “approaches sentence to paragraph level” while the maximum value of n seems to be 6 (which is shorter than most sentences, let alone paragraphs).

---

> ### Author Response · Authors · 2023-11-19
> **Response to Reviewer JMFi (Part 1)**
>
> **General Response:** Thank you for your review. We appreciate your recognition of the clarity of our paper. Your insights on the significance of document length and the consideration of multiple sampled documents are valuable and will further enrich the discussion in this paper.
>
>
> ### Weakness
>
> > Weaknesses 1: My main concern is that the results in this paper are general facts about sample complexity and not specific to the detection of AI-generated text. For example, Theorem 1 is included here: https://github.com/ccanonne/probabilitydistributiontoolbox/blob/master/testing.pdf. As this paper is outside of my area, I am less confident about the novelty of Theorem 2, but its formulation is not dependent on the domain of AI-generated text detection. It is also unclear from the paper whether the “multiple samples” required by this theorem are meant to be individual words, sentences, or documents.
>
> **Response to Weakness 1** We thank the reviewer for the reference but there seems to be a confusion regarding the novelty of our work. We note that the upper bound in the Total variation distance of the distribution (product distribution) is deep-rooted in the fundamentals of Information theory more specifically Lecam's lemma (as clearly highlighted at multiple places in our paper such as in the start of Section 3.2, before Equation (9),  and Appendix B.1). We never claim the novelty regarding the same, whereas ***our main novelty*** lies in deriving a precise sample complexity bound in the context of AI-generated text detection for both iid and non-iid scenarios, which is novel.
>
> ***Our Results are for AI-generated Text Detection (as acknowledged by [Reviewer 6pNs](https://openreview.net/forum?id=oxEER3kZ9M&noteId=9zofG5ezim)):*** We illustrate our results precisely in the context of LLMs through several instances, provide key insights and also empirical evaluation ***in the text domain***. For example in Section 3.4, we develop the ***novel connection between non-iid sample complexity and increasing sequence length in generation***, illustrate the implication of $\sum_k s_k$ (Equation 16) and novel insights on dependence of topics/contexts present in the paragraph with sample complexity results and empirical evaluations with several SOTA generators and detectors. We want to highlight that our results and comparisons are inline with the recent results in the field of AI-generated text detection (Sadasivan et. al, Krishna et. al, Mitchel et al etc.)
>
> ***Clarification Regarding the use of the term 'multiple samples':*** Our result does not depend on the specific choice of the unit (for samples) to represent i.e., it can be a bag of words, sentences, or paragraphs, but rather emphasizes on the impact of product distribution on the sample complexity. ***To highlight the point we empirically demonstrate*** with several such instances like bag of words (Figure 2), sentences (Figure 3, 4), etc.

---

> > ### Author Response · Authors · 2023-11-19
> > **Response to Reviewer JMFi (Part 2)**
> >
> > > Weakness 2 : The experiments also replicate known results, namely, that AI-generated text detectors perform better on longer sequences and that they perform worse with paraphrasing. As far as I can tell, the main novel finding is the “pairwise with IID samples” condition (Figure 2c). It would be interesting to see a more controlled experiment here, which compares the performance of detectors on two IID documents of lengths X,Y versus one single document of length X+Y.
> >
> >
> > **Response to Weakness 2** We ***respectfully disagree with the reviewer*** that our experiments replicate the known results. ***Our work is the first*** to show (with empirical experiments) a precise connection of the detection performance with the sample complexity (no. of samples (Theorem 1), length of sequence (Theorem 2)) in the context of LLMs, which has been a critical point of discussion in the recent times. We are aware of the concurrent works for this problem and cannot provide further details as it will violate anonymity. We will provide more information to the AC.
> >
> > Regarding the point on the controlled experiment, we believe we have the experimental results in similar settings for example in Figures 2b, 2c. Figure 2c (iid box plot) represents the scenario with pairwise samples (sequences) as input to the detector and Figure 2b represents the scenario with varying sequence lengths validating Theorem 1 and 2 respectively. It is evident that even with just pairwise samples, the detector performance is very good achieving an AUC of 95% which exactly validates our Theorem.
> >
> > ### Questions:
> >
> > > Question 1: Could you provide additional details on the process used to generate Figure 2a? How were documents tokenized and how was AUC computed? I am also confused by the statement in Section 4.1 that this analysis “approaches sentence to paragraph level” while the maximum value of n seems to be 6 (which is shorter than most sentences, let alone paragraphs).
> >
> > **Response to Question 1:** To generate Figure 2a, we discretize the sequences (paragraphs) from the Xsum or Squad dataset using bag of words features varying the ngrams from n = 1 to 6. For each of the scenarios, we compute the AUC of the best detector using equation 13 (similar to Sadasivan et. al Figure 7). It is important to note that in the finite discretized feature space, TV distance can be computed using the definition [1] and then we plot the AUC upper bounds by varying the value of n in n-grams as in Figure 2a. The key message (via Figure 2a) is to demonstrate the significant improvement in AUC with increasing n-grams validating our Theorems.
> >
> >
> > ***Regarding the statement in Section 4.1***, we meant to say that as the token length increases (word to sentence, and sentence to paragraph, and so on), it results in a significant increase in AUROC, hence better detectability as observed in Figure 2. Also, to clarify with ngrams = 6, it means considering bag of 6 words as each feature and this doesn't represent the number of features, which is different (num_features = 100). More specifically, it means that each feature is a collection of n = 6 words (almost like a sentence) and we can see a significant improvement in AUC of 95% (achieved by the best detector). We have also experimented with n = 7, 8, and 10 bag of words-based features and observed nearly 98% AUC achieved by the best detector. Hope this clears the confusion.
> >
> > We will clarify this and update the discussion (providing more details about n-grams and features) in the revised version of our paper.
> >
> > We believe that our responses have addressed your concerns. It is our pleasure to provide additional details if needed. If your concerns have been resolved, we kindly request you to consider increasing the scores.
> >
> > [1]. Bharath K. Sriperumbudur, Kenji Fukumizu, Arthur Gretton, Bernhard Schölkopf, Gert R. G. Lanckriet. On integral probability metrics, ϕ-divergences and binary classification, https://arxiv.org/abs/0901.2698

---

> > > ### Author Response · Authors · 2023-11-19
> > > **Request to the Reviewer**
> > >
> > > Dear Reviewer,
> > >
> > > Thank you so much for your time and efforts in reviewing our paper. We have addressed your comments in detail and are happy to discuss more if there are any additional concerns. We are looking forward to your feedback and would greatly appreciate you consider raising the scores.
> > >
> > > Thank you,
> > >
> > > Authors

---

> > > > ### Author Response · Authors · 2023-11-22
> > > > **Request to the Reviewer**
> > > >
> > > > Dear Reviewer,
> > > >
> > > > Since the deadline for the author-reviewer discussion period is approaching (today is the deadline), we wanted to humbly check with the reviewer for any remaining concerns. We are happy to discuss more. We are looking forward to your feedback.
> > > >
> > > > Thank you,
> > > >
> > > > Authors

---

### Official Review · Reviewer_Xeu7 · 2023-11-01

**Soundness:** 2 fair
**Presentation:** 3 good
**Contribution:** 3 good
**Rating:** 5
**Confidence:** 4

**Summary:**

This paper studies the possibility of detecting machine-generated texts. With a theoretical analysis of true positive rate/false positive rate, the high-level message is that detecting MGT is possible if we have sufficient examples from humans and the machine. Empirical study on four datasets with a wide range of combinations between generation models and detection models demonstrate the utility of the theoretical results.

**Strengths:**

- The research problem is important and very challenging
- The theoretical analysis is strong and sounding
- The results support the high-level conclusion, although I am not sure whether it directly supports the conclusion about sample complexity  (please refer to the question section)

**Weaknesses:**

- Interesting method and conclusion; however not sure how much we can connect this with MGT detection. It seems like the conclusion is generally applicable. This means the proposed method itself is not necessarily a weakness, but the disconnection is.
- Equation 7 is not surprising. As pointed out by Sadasivan et al. (2023), it is impossible to get a reliable detector (high TPR, e.g., 90% and lower FPR, e.g., 1%), when the overlap of two distributions is relatively small.
- Furthermore, equation 7 does not depend on sample complexity. So, I am not sure I understand how increasing the number of examples can get around this issue pointed out by the previous comment.
- More generally, the real challenge of detecting MGT is the lack of information on generative models. In practice, it’s hard to predict whether a collection of texts is from the same generation model. Furthermore, a gray area of this research question: how we should treat the texts generated by machines and edited by humans.

**Questions:**

- What is the “Percentage of Sequence used for Detection”?
- What does “exponentially fast” mean in the caption of Figure 1?
- Are there any experiment results on non-IID data?
- Maybe I missed something from the paper, but is there any sample complexity directly related to equation 15?

---

> ### Author Response · Authors · 2023-11-17
> **Response to Reviewer Xeu7 (Part 1)**
>
> **General Response:** We thank the reviewer for appreciating the research problem and theoretical analysis. We address the additional concerns in detail as follows.
>
> ### Weaknesses:
>
> > Weakness 1: Interesting method and conclusion; however not sure how much we can connect this with MGT detection. It seems like the conclusion is generally applicable. This means the proposed method itself is not necessarily a weakness, but the disconnection is.
>
>
> **Response to Weakness 1** We are thankful that you find our formulation and methodology interesting.
>
> Regarding the generality of the results, we agree that certain theoretical results for example used in Eq 6, 7, and 9 have their roots in Information theory and Large deviation theory ***but a precise connection to the context of AI-generated text detection was missing from literature.*** Specifically, our sample complexity analysis for iid samples (Theorem 1) and non-iid samples (Theorem 2) are novel and provide a ***first step towards showing possibility*** results (in a theoretically rigorous manner) of MGT detection.
>
> We illustrate our results precisely in the context of LLMs through several instances, key insights, and empirical evaluation. For example in Section 3.4, we develop the connection between non-iid sample complexity and increasing sequence length in the generation, illustrate the implication of $\sum_k s_k$ (Equation 16), and novel insights on the dependence of topics/contexts present in the paragraph with sample complexity results and empirical evaluations with several SOTA generators and detectors.
>
> > Weakness 2: Equation 7 is not surprising. As pointed out by Sadasivan et al. (2023), it is impossible to get a reliable detector (high TPR, e.g., 90% and lower FPR, e.g., 1%), when the overlap of two distributions is relatively small.
>
> **Response to Weakness 2** We agree that Equation 7 trivially follows and it comes from an application of Lecam's Lemma from Information Theory which we have clearly cited (and not first appeared in Sadasivan et al. (2023)). But we respectfully point out that there seems to be a slight confusion here. Our point is not about Equation (7). We would like to highlight the **hidden possibility** part which starts around Equation (8) (which is different from Sadasivan et al. (2023)).
>
> **Our Focus:** Our result in this paper focuses on Equation (9), which helps to conclude that even if the overlap of distribution in Equation (7), is small, increasing the samples/sequence length helps to increase the TPR because TV norm increases exponentially with respect to $n$ as highlighted in (12). This is rigorously connected with bounds developed in Theorem 1 (iid) and Theorem 2 (non-iid) cases.
>
> > Weakness 3: Furthermore, equation 7 does not depend on sample complexity. So, I am not sure I understand how increasing the number of examples can get around this issue pointed out by the previous comment.
>
> **Response to Weakness 3:** Equation 7 is mentioned to lay the foundation of our discussion but our results follow from the expression in Equation (9) which clearly depends upon the values of $n$ (samples/sequence length). We apologize for the confusion, we will highlight this fact in the revised version of our manuscript.
>
> > Weakness 4: More generally, the real challenge of detecting MGT is the lack of information on generative models. In practice, it’s hard to predict whether a collection of texts is from the same generation model. Furthermore, a gray area of this research question: how we should treat the texts generated by machines and edited by humans.
>
> **Response to Weakness 4** This is a very interesting point. We agree that texts generated by machines and edited by humans is a very challenging setting and is very hard to detect in that case. However, ***our result provides a first step to rigorously show the possibility of detection*** where the text comes from either human/machine distribution. However, texts generated by machines and edited by humans are an extremely important and valid source of future research.

---

> ### Author Response · Authors · 2023-11-17
> **Response to Reviewer Xeu7 (Part 2)**
>
> ### Questions
>
> > Question 1. What is the “Percentage of Sequence used for Detection”?
>
> **Response to Question 1** We vary the sequence length from 10% to 95% as shown in Figures 3a-f, and it is evident the detection performance improves significantly.
>
> > Question 2: What does “exponentially fast” mean in the caption of Figure 1?
>
> **Response to Question 2** In Figure 1, the value of Total variation distance (best detector) increases at an exponential rate as a function of the number of samples, which is also evident from our theoretical analysis as shown in Equation 12. In equation 12, it is evident that as we increase n, TV exponentially approaches 1.
>
> > Question 3: Are there any experiment results on non-IID data?
>
> **Response to Question 3:** ***Yes, We want to highlight that all our experimental*** ablations including Figures 2 (a,b) 3 (a->f) ***are for non-iid scenarios only***, as you can see that the performance (AUC) is measured with increasing sequence length. This is exactly what we discuss in Theorem 2, when we increase the sequence length (which is equivalent to having more dependent samples) the detection performance varies, which is novel and not covered in past literature.
> We apologize if this point didn't come out clearly and will be updating the draft with a specific focus on this point as also highlighted by Reviewer 6pNs: "I feel like the authors are under-selling themselves in the title/abstract/introduction, by focusing on the "number of samples" rather than "sequence length"
>
> > Question 4: Maybe I missed something from the paper, but is there any sample complexity directly related to equation 15?
>
> **Response to Question 4** Equation 15 represents the sample complexity results for the iid case (Theorem 1) which states that to achieve an AUROC of $\epsilon$, we will need $O(\frac{1}{\delta^2} \log \frac{1}{1 - \epsilon})$ samples (iid case, non-iid in Theorem 2). This precisely represents the sample complexity of detection i.e how the detection performance varies (in terms of AUC) with number of samples or sequence length. Hope this clears the confusion.

---

> > ### Author Response · Authors · 2023-11-19
> > **Request to the Reviewer**
> >
> > Dear Reviewer,
> >
> > Thank you so much for your time and efforts in reviewing our paper. We have addressed your comments in detail and are happy to discuss more if there are any additional concerns. We are looking forward to your feedback and would greatly appreciate you consider raising the scores.
> >
> > Thank you,
> >
> > Authors

---

> > > ### Author Response · Authors · 2023-11-22
> > > **Request to the Reviewer**
> > >
> > > Dear Reviewer,
> > >
> > > Since the deadline for the author-reviewer discussion period is approaching (today is the deadline), we wanted to humbly check with the reviewer for any remaining concerns. We are happy to discuss more. We are looking forward to your feedback.
> > >
> > > Thank you,
> > >
> > > Authors

---

> ### Author Response · Authors · 2023-11-23
> **New Experimental Results**
>
> **New Experimental Results ([Figure 11 in Appendix E in the updated draft](https://openreview.net/pdf?id=oxEER3kZ9M)):** We would like to provide an update that we have also validated the detection performance on Question-answering style task with OpenAI's Roberta classifier and observe that with the increase in sequence length for detection, the zero-shot detection performance of both the models improves significantly from around 55\% to 98\% validating our theorem for QA tasks.
>
> Please let us know if we can address any more concerns before the rebuttal period ends. Thank you so much. We look forward to your feedback.

---

### Official Review · Reviewer_hkaL · 2023-11-09

**Soundness:** 3 good
**Presentation:** 4 excellent
**Contribution:** 3 good
**Rating:** 5
**Confidence:** 2

**Summary:**

This work tackles the possibility of the detection of AI-generated texts from a perspective of increasing sample sizes (length). Specifically, the authors argue that there is a hidden possibility of the detection even with the machine and human text distributions close to each other, if more samples can be collected/used. They derive sample complexity bounds for their finding and corroborate the results with empirical experiments in conditional generation (e.g., on news or Wikipedia articles).

**Strengths:**

This paper is tackling an important problem on the possibility of detecting machine-generated text. The writing of the paper is overall clear, and the authors explore several empirical experiments to support the theoretical findings (on the hidden possibility of detection).

**Weaknesses:**

The novelty of the finding. While the "hidden possibility" argument over the increasing sample size and increasing detectability of machine-generated texts is valid and intuitive, the novel insight in the argument is rather limited. The assumption and use of an increasing sample size/length in the machine-generated text detection problem is regular, for example, in watermarking (Kirchenbauer et al., 2023). The practical takeaway of the finding is relatively vague.

The experimental setup. In (1) and (2) of Section 4.1, the detector setup is very basic, for example, with n-gram features and linear classifiers. Then, if a 6-gram detector (in the basic setup) can achieve 97% accuracy in detection, is the possibility of detection still a relevant concern? Additionally, the experiments can be conducted on question-answering/instruction-following based tasks (e.g., generating chatbot-style answers) instead of only performing completions on news or Wikipedia articles.

**Questions:**

Please see the weaknesses section.

---

> ### Author Response · Authors · 2023-11-17
> **Response to Reviewer hkaL (Part1)**
>
> **General Response:** Thank you for your review and recognition of the importance of our work in detecting AI-generated texts. We are glad you found the paper clear and the empirical experiments supportive of our theoretical findings. Your feedback is invaluable in highlighting the strength and importance of our research.
>
> ## Weakness
>
> > **Weakness 1:** The novelty of the finding. While the "hidden possibility" argument over the increasing sample size and increasing detectability of machine-generated texts is valid and intuitive, the novel insight in the argument is rather limited. The assumption and use of an increasing sample size/length in the machine-generated text detection problem is regular, for example, in watermarking (Kirchenbauer et al., 2023). The practical takeaway of the finding is relatively vague.
>
> **Response to Weakness 1**: Thanks for appreciating our finding regarding the "hidden possibility" of AI-generated text detection. Regarding the novelty, we want to point out that ***our work is the first attempt to theoretically show the "hidden possibility" of AI-generated text detection*** and derive the precise sample complexity analysis for the detection problem. These theoretical results provide the insight that the detectability increases as the ***number of samples (tokens)(Theorem 1)/sequence length (Theorem 2)*** increases and also derive the rate of increment. The referred work by Kirchenbauer et al., 2023 is a concurrent work (can't provide more details on the timeline due to anonymity violation) and provides no theoretical analysis, rather provides empirical demonstrations that even validate our theorems for iid and non-iid cases.
>
> ***The primary takeaway*** of our approach is to provide theoretically rigorous evidence (connecting it with the text sequence length) that the detection of AI-generated text detection is possible, which is in contrast to recently developed impossibility results in Sadasivan et. al. As highlighted by the Reviewer 6pNs: *"I feel like the authors are under-selling themselves in the title/abstract/introduction, by focusing on the "number of samples" rather than "sequence length"*, we will specifically highlight this aspect in our revised version of our work.
>
> **Importance of Our Results.** Additionally, we want to highlight that our theoretical results of characterizing the sample complexity with Chernoff information (for iid and non-iid scenarios) are critical and provide additional insights (detailed in Appendix A) for example
>
> 1. Watermark Design: Our analysis suggests adaptive watermark designs based on Chernoff information and sample complexity, enhancing robustness against attacks—a future research avenue.
> 2. Length Constraints in Exams: Our bounds inform optimal summary lengths for easier cheating detection, leveraging the Chernoff metric.

---

> > ### Author Response · Authors · 2023-11-17
> > **Response to Reviewer hkaL (Part2)**
> >
> > > **Weakness 2:** The experimental setup. In (1) and (2) of Section 4.1, the detector setup is very basic, for example, with n-gram features and linear classifiers. Then, if a 6-gram detector (in the basic setup) can achieve 97% accuracy in detection, is the possibility of detection still a relevant concern?
> >
> > **Response to Weakness 2** : Thanks for the question. We concede that the setup in (1,2) i.e Figure 2a,2b is basic with n-gram features and vanilla classifiers (logistic, random forest etc.) but that is an intentional choice. The primary purpose of these plots is to ***show a proof of concept*** of our theorems and demonstrate the improvement in AUC with ***increasing sequence length***. Specifically, we wanted to highlight the case that if the detection is done with lower n-grams (2-3 ngrams plot 2a), the detection AUC is around 50% and prior theory (Sadasivan et. al) results conclude that the detection is impossible.  In contrast, with a slight increase in n-grams to 4, the AUC shoots up making the detection possible highlighting our claims.
> >
> > However, we have performed detailed experiments on standard benchmarks (as used in Sadasivan et. al, Mitchell et. al, Krishna et. al etc.) on a variety of state-of-the-art text generators and detectors (Figure 3, 4, 9, 10) which validates our claims.
> >
> >
> > > **Weakness 3:** Additionally, the experiments can be conducted on question-answering/instruction-following based tasks (e.g., generating chatbot-style answers) instead of only performing completions on news or Wikipedia articles.
> >
> > **Response to Weakness 3:** Thank you for the valuable suggestion to expand our experimental framework to include question-answering and instruction-following tasks, such as generating chatbot-style responses. This would indeed provide a more comprehensive understanding of our model's capabilities, especially in more interactive and dynamic scenarios, compared to the current focus on news and Wikipedia article completions.
> >
> > We are running more experiments and will be including them in the final version of our work. However, we would also like to remark that the fundamental insights and conclusions of our research would remain unchanged even with more experiments.

---

> > > ### Author Response · Authors · 2023-11-19
> > > **Request to the Reviewer**
> > >
> > > Dear Reviewer,
> > >
> > > Thank you so much for your time and efforts in reviewing our paper. We have addressed your comments in detail and are happy to discuss more if there are any additional concerns. We are looking forward to your feedback and would greatly appreciate you consider raising the scores.
> > >
> > > Thank you,
> > >
> > > Authors

---

> > > > ### Author Response · Authors · 2023-11-22
> > > >
> > > > Dear Reviewer,
> > > >
> > > > Since the deadline for the author-reviewer discussion period is approaching (today is the deadline), we wanted to humbly check with the reviewer for any remaining concerns. We are happy to discuss more. We are looking forward to your feedback.
> > > >
> > > > Thank you,
> > > >
> > > > Authors

---

> ### Author Response · Authors · 2023-11-23
> **New Experimental Results on Question-Answering Based Task**
>
> **New Experimental Results ([Figure 11 in Appendix E in the updated draft](https://openreview.net/pdf?id=oxEER3kZ9M)):** As requested by the reviewer, we have also validated the detection performance on Question-answering style task with OpenAI's Roberta classifier and observe that with the increase in sequence length for detection, the zero-shot detection performance of both the models improves significantly from around 55\% to 98\% validating our theorem for QA tasks.
>
> Please let us know if we can address any more concerns before the rebuttal period ends. Thank you so much. We look forward to your feedback.

---

### Meta-Review · Area_Chair_Afh6 · 2023-12-12

**Metareview:**

This paper theoretically justifies that detecting whether or not a piece of text is written by an AI is feasible with multiple samples from the AI and longer outputs. The paper partially rebuts the strongly phrased "impossibility result" of Sadasivan et al, demonstrating more hope for AI-generated text detection. However, reviewers were concerned by several aspects of the paper. First, the findings are not particularly surprising: it is well known that detection rate increases with length, and it is similarly intuitive that it would increase with multiple samples. Second, the emphasis on multiple sample detection is impractical in most real-world scenarios, in which only one sample will be available. Even in the Twitter bot setting, there is no guarantee that the bot is using the same model for each fake post. Reviewer 6pNs correctly suggests that the paper be reframed to focus more on sequence length rather than multiple samples; in general, several reviewers had issues understanding what was even meant by "multiple samples". Overall, it is clear that there is something interesting in this paper, but it needs substantial rewriting to improve its clarity before it is worthy of acceptance.

**Justification For Why Not Higher Score:**

The paper lacks clarity and needs significant rewriting to highlight its real contributions to practical research on LLM-generated text detectors.

**Justification For Why Not Lower Score:**

N/A

---

### Decision · Program_Chairs · 2024-01-16

Reject